

**Development of flexible double distribution quantile mapping for better**
**bias correction in precipitation of GCMs**
Young Hoon Song[1], Eun-Sung Chung[1*], Shamsuddin Shahid[2]
[1] Faculty of Civil Engineering, Seoul National University of Science and Technology, 232
Gongneung-ro, Nowon-gu, Seoul 01811, Korea
[2] School of Civil Engineering, Faculty of Engineering, Universiti Teknologi Malaysia (UTM),
81310 Johor Bahru, Malaysia
[*] Correspondence to: Eun-Sung Chung eschung@seoultech.ac.kr
**Abstract**
The double gamma quantile mapping (DGQM) can outperform single gamma quantile
mapping (SGQM) for bias correction of global circulation models (GCMs) using two gamma
functions for two segments based on 90[th] quantile. However, there are two ambiguous points:
the 90[th] quantile and considering only the Gamma probability function. Therefore, this study
introduced a flexible dividing point, $\delta$ (%), which can be adjusted to the regionally observed
values at the station and considered the combination of various probability distributions,
Weibull, lognormal, and Gamma, for two separate segments. The newly proposed method,
flexible double distribution quantile mapping (F-DDQM), was employed to correct the bias of
8 GCMs of Coupled Model Intercomparison Project Phase 6 (CMIP6) to correct bias at 22
stations in South Korea. The results clearly showed higher performance of F-DDQM than
DGQM and Flexible-DGQM (F-DGQM) by 25% and 5%, respectively, in root mean square
error. The F-DGQM also showed better performance in replicating probability distribution,
spatial variability and extremes of observed precipitation than other methods. This study
contributes to improving the bias correction method for the better projection of extreme values.
**Keywords**: Double gamma quantile mapping, Bias correction method, Flexible double gamma
quantile mapping, Flexible double distribution quantile mapping





## 1. Introduction

Global circulation models (GCM) provide insight into the historical and possible future climate variabilities and the occurrence of extreme events (Ahmed et al., 2018; Pour et al., 2018). Therefore, climate studies generally use GCMs to simulate historical and future climate conditions (Shiru et al., 2022; Song et al., 2022c; Iqbal et al., 2020). The reliable simulation of precipitation is important for climatological and hydrological science. However, the GCMs outputs have biases in the simulation due to imperfect model parameterization, inadequate reference data, and incomplete knowledge (Wilby and Harris 2006; Woldemeskel et al., 2014). Besides, the previous studies showed that raw GCMs can not replicate the observed climate of South Korea due to its complicated geographical characteristics (Song et al., 2021a). Therefore, various bias correction techniques have been used to correct the bias in GCM simulations before their use for climatic studies.

The distribution-derived transformations, such as quantile mapping (QM), are most widely used for bias corrections because of their simplicity and easy employability but higher proficiency (Ringard et al., 2017; Maraun et al., 2010; Ines and Hansen, 2006; Li et al., 2010). The QM shows high performance in bias correcting stationary climate variables but low reliability for nonstationary data. Cannon et al. (2015) proposed a quantile delta mapping (QDM) method to preserve the relative change in all quantiles to address the nonstationary issue. Several methods have been developed in recent years based on QDM to enhance the bias-correction ability, including scaled distribution mapping (Switanek et al., 2017), multivariate quantile delta mapping (Cannon, 2017), and the occurrence-and intensity bias-adjusting methods (Van de Velde et al., 2020).

The QM method replaces the quantiles of simulated data corresponding to a given probability and the observed quantile corresponding to the same probability (Cannon, 2008; Piani et al., 2010; Cannon, 2012; Heo et al., 2019). The QM uses different probability distributions for this purpose, such as Gamma, Weibull, and exponential. Besides, Ye et al. (2018) suggested the three-parameter Gamma distribution. Nevertheless, QM does not always outperform other bias-correction methods at all locations (Song et al., 2020). This emphasizes choosing an appropriate probability distribution function (PDF) for successful bias correction.

In general, the gamma distribution is used in QM. The gamma quantile mapping (GQM) inflates the extreme precipitations (Cloke et al., 2013, Huang et al., 2014). Several studies have demonstrated that GQM underestimates the extremes which affects the design precipitation


(Hundecha et al., 2009; Volosciuk et al., 2017; Vrac and Naveau, 2007; Kim et al., 2018). Yang
et al. (2015) proposed double gamma quantile mapping (DGQM) to efficiently correct the
biases in extreme precipitation, which has demonstrated superior performance to single GQM.
Pasten-Zapata et al. (2020) showed that the bias performance of DGQM is higher than single
GQM. In DGQM, the fixed value, 90th quantile, is popularly used to divide the entire data set
into two segments for two separate GQMs. However, the 90th is not always accurate in
estimating precipitation extremes at all locations because, theoretically, this value is not fixed.
In addition, the gamma distribution function is popularly used for the bias correction in
precipitation. However, the most appropriate distribution can be different for different regions.
This indicates the need for selecting appropriate probability distribution based on study
location to improve the performance of the bias correction method.
This study proposed a new flexible double distribution quantile mapping (F-DDQM) method
considering adjustable dividing points and two individually selected distributions for two
segments. Three PDFs, Weibull, lognormal, and Gamma distributions, were considered for
selecting appropriate PDF for two segments. The dividing point was determined based on the
optimal RMSE of the overall precipitation distribution. The proposed method was employed
for correcting the bias of 8 GCMs of Coupled Model Intercomparison Project 6 (CMIP6) at 22
stations in South Korea. The performance of the proposed was compared with the DGQM and
the Flexible DGQM (F-DGQM) using five evaluation metrics to show its efficacy. Furthermore,
the performance of the proposed method in correcting the bias of extreme precipitation based
on GEV distribution. Besides, the difference between the simulated precipitation distribution
and the observed distribution was compared using Jensen-Shannon (JSD) and Kullback-Leibler
divergence (KLD). This study contributes to improving the bias correction method for the
better projection of extremes.

**2. Study area and data**
**2.1 Study area**
South Korea, located in Asia, lies between Japan and China. The country has four distinct
seasons: winter (DJF), spring (MAM), summer (JJA), and autumn (SON). South Korea has
mountainous topography in more than half of its total area. Therefore, the climate varies
significantly among regions due to large topographical variability. The annual average





precipitation ranges between 1000 mm and 1600 mm. The majority of precipitation occurs in
summer.
**2.2 Dataset and sources**
This study used monthly precipitation simulations of 8 GCMs of CMIP6, as listed in Table 1.
The resolutions of the GCMs range from 0.98° to 2.81°. The CMIP6 GCMs selected in this
study are frequently used in East Asia, including South Korea climate studies (China: Wu et
al., 2020; Yue et al., 2021; Lun et al., 2021, South Korea: Song et al., 2021b; Kim et al., 2021;
Chae et al., 2022). Some studies also evaluated the performance of these GCMs in South Korea
(Song et al., 2020). The CMIP6 GCMs outputs were collected from data portals (https://esgf-
node.llnl.gov/search/cmip6/).
The monthly precipitation of 22 gauges was used in this study (Figure 1). They were selected
from 96 gauges available in South Korea, considering the availability of monthly rainfall
records without missing data for the historical period (1970-2014). The selected stations are
exposed to several hydrological disasters, such as floods and heavy snow. Therefore, the high
reproducibility of precipitation can improve the accuracy of precipitation projections when
analyzing disasters due to precipitation changes in South Korea.

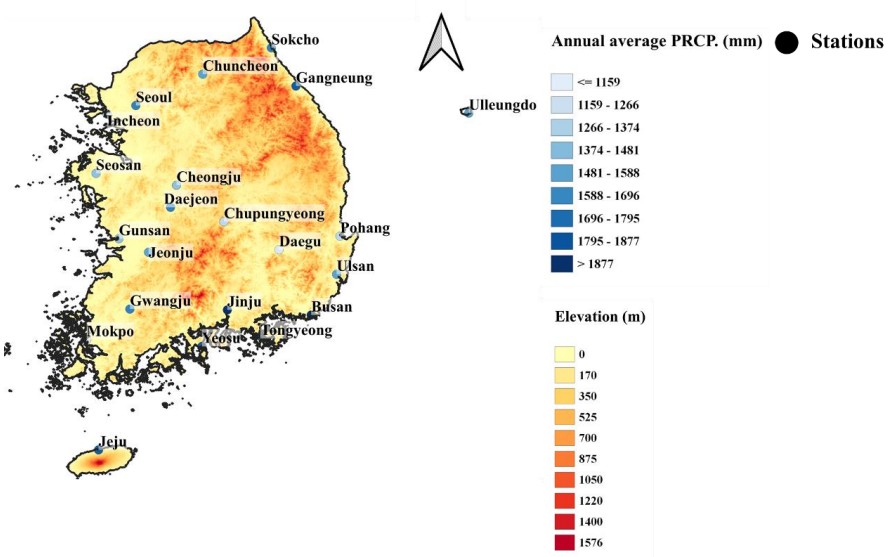


Figure 1. Location of the selected stations in South Korea.






Table 1. Information about GCMs used in this study.

| Institute | Models | Resolution (Longitude × Latitude) |
|---|---|---|
| Commonwealth Scientific and Industrial Research Organisation, and Bureau of Meteorology | ACCESS-ESM1-5 | 1.25° × 1.875° |
| Canadian Earth System Model version 5, Canadian Centre for Climate Modelling and Analysis (Canada | CanESM5 | 2.81° × 2.81° |
| NASA Goddard Institute for Space Studies | GISS-E2-1-G | 2.0° × 2.5° |
| Institute for Numerical Mathematics, Russian Academy of Science, (Russia) | INM-CM4-8 | 1.5° × 2.0° |
| Institut Pierre Simon Laplace | IPSL-CM6A-LR | 2.5° × 1.26° |
| Max Planck Institute for Meteorology (MPI-M) (Germany) | MPI-ESM1-2-LR | 1.125° × 1.12° |
| Meteorological Research Institute (Japan) | MRI-ESM2-0 | 1.125° × 1.125° |
| Norwegian Climate Centre (Norway) | NorESM2-MM | 1.25° × 0.9375° |


**3. Methodology**
**3.1 Inverse distance weighted method**
The CMIP6 GCMs outputs are in the form of a grid with fixed resolution. The geographical
interpolation methods are used to remove the spatial difference between the GCM simulation
and the observed data. The inverse distance weighted (IDW) method has been widely used for
geographical interpolation (Longley et al., 2005). The concept of IDW is based on Tobler's first
law, in which data from the nearby point are more relevant than distant point (Tobler, 1970).
Equation 1 is used to estimate the CMIP6 GCM precipitation at the observed locations from
their values in nearby locations. Equation 2 computes the interpolation weight for the distance
between the grid and the interpolation points.
$P_i = \sum_{k=1}^{N} \frac{w_s(x)}{\sum_{k=1}^{N} w_s(x)} P_i(x_s)$                    (1)



$w_s(x) = \frac{1}{D^c_{(x,x_s)}}$ (2)
where $P_i$ is the precipitation in the interpolation area, $P_i(x_s)$ is the GCM precipitation at
grids surrounding the observed location, $w_s$ is the interpolation weight, and $D_{(x,x_s)}$ is the
distance between the interpolation and grid point. This study used the Shepard method to
estimate the interpolation weight, and the pattern is interpolated narrowly ($0 < D^c < 1$) or
widely ($D^c > 1$) depending on $D^c$. This study used 50 grids close to 22 stations for spatial
downscaling.

**3.2 Single & Double gamma quantile mapping**
Distribution-derive transformations of QM are bias-correction techniques depending on
distribution parameters. These methods use distribution functions, such as Gamma, Lognormal,
and Weibull, to reduce the differences between the observed and GCM raw data (Piani et al.,
2010). The single gamma quantile mapping (SGQM) is most widely used to reduce the
differences between GCM outputs and observed data using their cumulative distribution
function (CDF), as shown in Equation 3.

$P_o(t) = F_g^{-1}\big(F_g(P_m(t), \alpha_m, \beta_m), \alpha_o, \beta_o\big)$ (3)

where $P_o(t)$ denotes the bias-corrected monthly precipitation, $P_m(t)$ represents GCM raw
data, $F_g^{-1}$ is the inverse CDF of the observed data to which the gamma function is applied,
and $F_g$ is the CDF of the GCM outputs. $\alpha_o$, $\alpha_m$, $\beta_o$ and $\beta_m$ represent shape and scale
parameters of observed and GCM simulation, respectively.
The SGQM tends to be more inflated than the observed data. Therefore, some studies used
double gamma quantile mapping (DGQM) for bias correction. DGQM is similar in
methodology to SGQM, with the difference being the division of the simulated precipitation
distribution into two segments by $\delta$. However, the bias correction is performed by randomly
determining the criterion of δ in most studies (Pastén-Zapata et al., 2020; Meresa et al., 2021).
Therefore, this study proposed a double distribution bias correction that can flexibly use δ.

**3.3 Flexible double gamma quantile** mapping (F-DGQM)





The F-DGQM is similar to the methodology of DGQM but uses δ to separate the two segments
flexibly. Figure 2 shows the concept of F-DGQM. The upper δ, representing quantiles between
80% and 95%, is selected based on the optimal root mean square error (RMSE). The upper δ
is determine based on optimal RMSE of the distributions of quantiles among 80–95%.

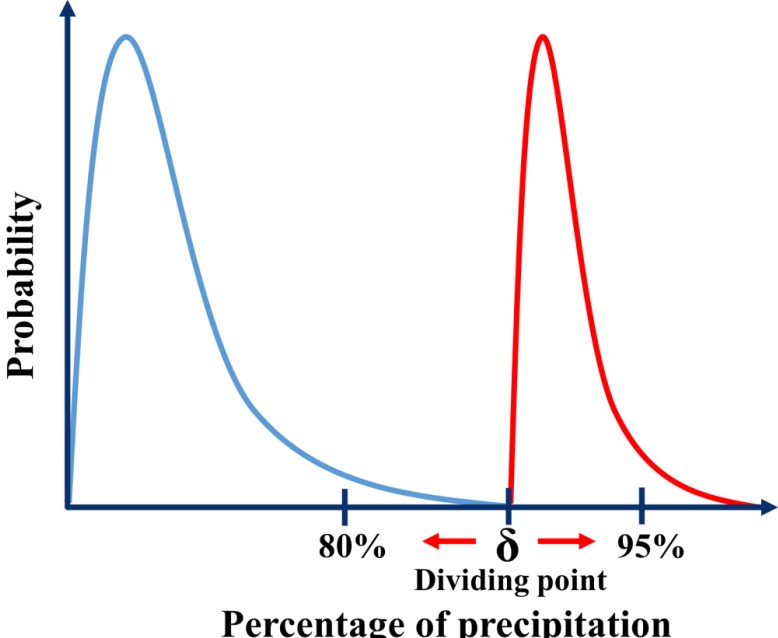


Figure 2. Concept of flexible double gamma quantile mapping (F-DGQM) based on optimal
RMSE

**3.4 Flexible double distribution quantile mapping**
The gamma distribution may not be appropriate for all observed data. Indeed, some studies
have argued that other distributions perform better than Gamma distribution (Gudmundsson et
al., 2012). Therefore, this study proposed determining the appropriate distribution for upper δ
and lower δ based on the RMSE from three distribution functions, Weibull, Lognormal, and
Gamma. The F-DDQM selects the optimal distribution for each segment after determining δ
based on the optimal RMSE, as shown in Figure 3.

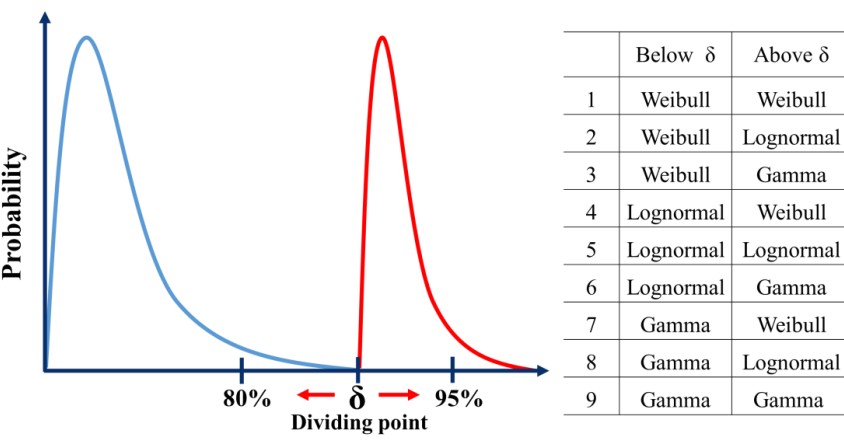


Figure 3. Concept of flexible double distribution quantile mapping (F-DDQM) based on RMSE


The proposed method can be used for bias correction of various climate variables. However, since the natural variability of precipitation is higher than the other climate variables, this study considered only precipitation bias correction (Deser et al., 2012; Cannon et al., 2015).


**3.5 Evaluation metrics**

This study used five evaluation metrics to evaluate bias corrected monthly precipitation performance using four distribution quantile mapping methods. The evaluation metrics used are as follows: normalized root mean square error (NRMSE), the percent bias (Pbias), the Nash-Sutcliffe efficiency (NSE) (Nash and Sutcliffe, 1970), a modified index of agreement (MD) (Willmott, 2013), and the Kling-Gupta efficiency (KGE) (Gupta et al., 2009). The evaluation metrics in this study are presented in Equations 4-8. In all equations, $X_s$ is the GCM outputs, $X_o$ is the observed data.

$$\text{NRMSE} = \frac{\sqrt{\frac{1}{n}\sum_{i=1}^{n}(X_s - X_o)^2}}{\overline{X_o}} \qquad (4)$$

The NRMSE is the result after removing the scale of RMSE. The values closer to 0 indicate higher accuracy.

$$\text{Pbias} = \frac{\sum_{i=1}^{n}(X_o - X_s)}{\sum_{i=1}^{n}X_o} \qquad (5)$$



Pbias represents the bias in the GCM and observation values. The tendency of overestimation
indicated positive value and vice versa.
$$NSE = 1 - \frac{\sum_{i=1}^{n}(X_s - X_o)^2}{\sum_{i=1}^{n}(X_o - \overline{X_o})^2} \tag{6}$$
NSE determines the relative magnitude of the residual variance in GCM simulations compared
to the variance in the station observation (Nash and Sutcliffe, 1970).
$$MD = \frac{[1 - (\sum_{i=1}^{n} abs(X_o - X_s))]}{(\sum_{i=1}^{n} abs(X_s - \overline{X_o})) + (abs(X_o - \overline{X_o}))} \tag{7}$$
MD estimates the sum and proportional difference between the observed and GCM data
(Willmott, 2013).
$$KGE = 1 - \sqrt{(1-r)^2 + (1-\alpha)^2(1-\beta)^2} \tag{8}$$
KGE is an integrated statistical metric that merges correlations, biases, and variability to assess
associations and errors in the mean and variability of the observed and GCM simulated data.
The optimal value is close to 1 (Gupta et al., 2009).

### 207    3.6 Generalized extreme value

The L-moment better estimates the GEV parameters than the maximum likelihood estimation,
particularly for small data samples (Hosking 1985). Since this study used 45 years of monthly
precipitation, the GEV parameters were estimated using the L-moment method (Hosking 1990).
The CDF of the GEV distribution is given in Equation 9.
$$G(x) = \left\{ -\left[1 - \frac{k(x-\xi)}{a}\right]^{1/k} \right\} \tag{9}$$
where $\xi, a, k$ are the location, scale, and shape parameters, respectively. The GEV combines
three probability distributions: Fréchet, Weibull, and Gumbel, with different representations of
the distributions depending on the value of the location ($\xi < 0$ is a Weibull; $\xi = 0$ is a Gumbel,
and $\xi > 0$ is a Frechet). The GEV corresponds to type I, II, and III, respectively, when the $a$
equals 0, greater than 0, and lower than 0 (Coles et al., 2001). This study compared the extreme
values of monthly precipitations bias-corrected using four QM methods considering GEV
distribution.

### 221    3.7 Kullback–Leibler & Jensen-Shannon divergence



KLD estimates the difference between PDFs based on their relative entropy (Kullback and
Leibler 1951). In other words, it estimates how well the model's simulation values preserve the
amount of information about the observed data. Equation 10 represents the expected value of
the amount of information loss using KLD.
$KLD(P\|Q) = \int_{-\infty}^{\infty} P(x) \log \frac{P(x)}{Q(x)} dx$ \hfill (10)
where $P(x)$ and $Q(x)$ are the continuous PDF of observed data and model simulation,
respectively, depending on distribution type. KLD is not symmetric for different probability
distributions.
JSD estimates the symmetric relationship and the distance between PDFs (Lin, 1991), as shown
in Equation 11.
$JSD(P,Q) = \frac{1}{2} D_{KL}(P\|\frac{P+Q}{2}) + \frac{1}{2} D_{KL}(Q\|\frac{P+Q}{2})$ \hfill (11)
This study compared the difference between the PDFs of observed and the bias-corrected
precipitation using KLD and JSD.

**4. Results**
**4.1 Flexible double gamma quantile mapping**
**4.1.1 Estimation results for $\delta$**
In this study, the $\delta$ of DGQM was determined according to optimum RMSE. Table S1
presents the estimated $\delta$ of F-DGQM based on the RMSE at 22 stations. Overall, most GCMs
showed the highest RMSE at the 80[th] quantile at 22 stations, except IPSL-CM6A-LR and MRI-
ESM2.0. IPSL-CM6A-LR showed the highest RMSE at 86[th] quantile and MRI-ESM2-0 at 93[rd]
quantile at 22 stations. This study compared the RMSE of 8 CMIP6 GCMs depending on the
change in $\delta$. Figure 4 presents the calculated RMSE according to $\delta$ at Seoul station. The most
selected quantile was the 80th, followed by the 90[th] at the Seoul station.



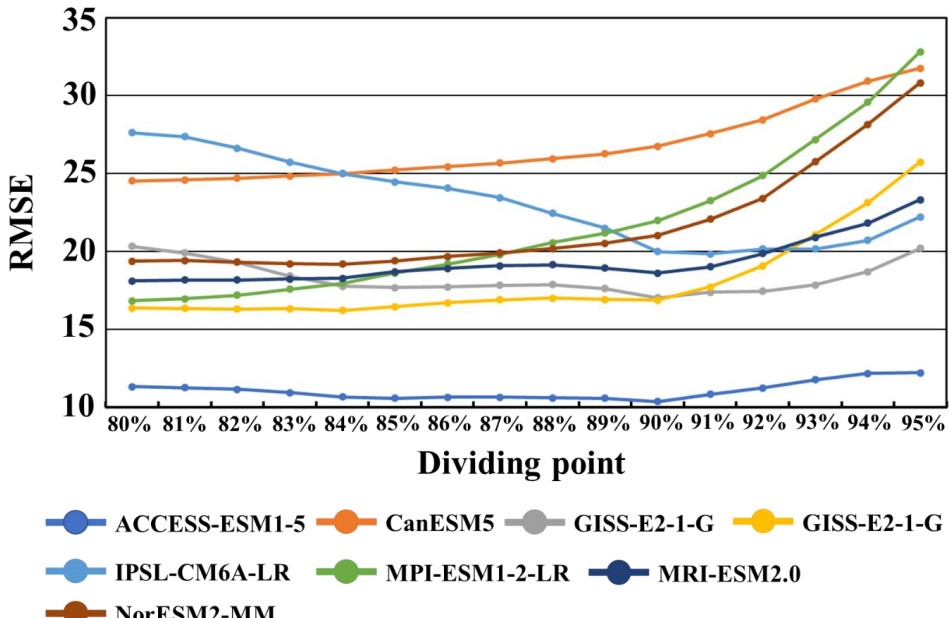

Figure 4. Comparison of RMSE of 8 CMIP6 GCMs depending on δ at the Seoul station.

The δ selected at 22 stations is presented using a heatmap in Figure 5. Overall, the selected δ was 80th quantile at most stations, followed by 95th quantile. The lowest selected quantiles were between 87th and 89th. Therefore, the appropriate δ was selected at both extremes of each quantile, whereas the 89-91% for δ was the opposite.



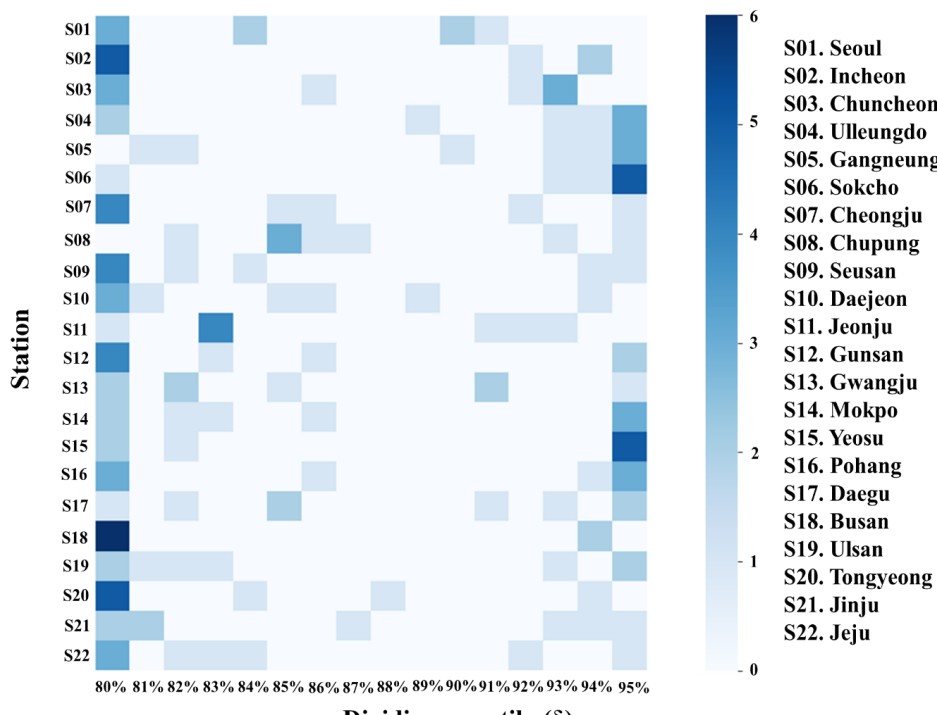

Figure 5. The heatmap shows the number of selected δ for F-DGQM depending on RMSE at 22 stations

## 4.1.2 Evaluation of results

This study compared the performance of F-DGQM with SGQM and DGQM. Figure 6 presents the methods' performances at 22 gages using box plots. The NRMSE for F-DGQM was the lowest (median < 0.1) among the QM methods, and the median NRMSE of F-DGQM was calculated below 0.1. The medians of F-DGQM Pbias was closer to the optimal value, whereas the SGQM overestimated and DGQM underestimated the observation. The median NSE of F-DGQM was higher than those for SGQM and DGQM. Besides, the median MD and KGE of F-DGQM were close to the optimum value. The results indicate better performance of F-DGQM than DGQM in all evaluation metrics.

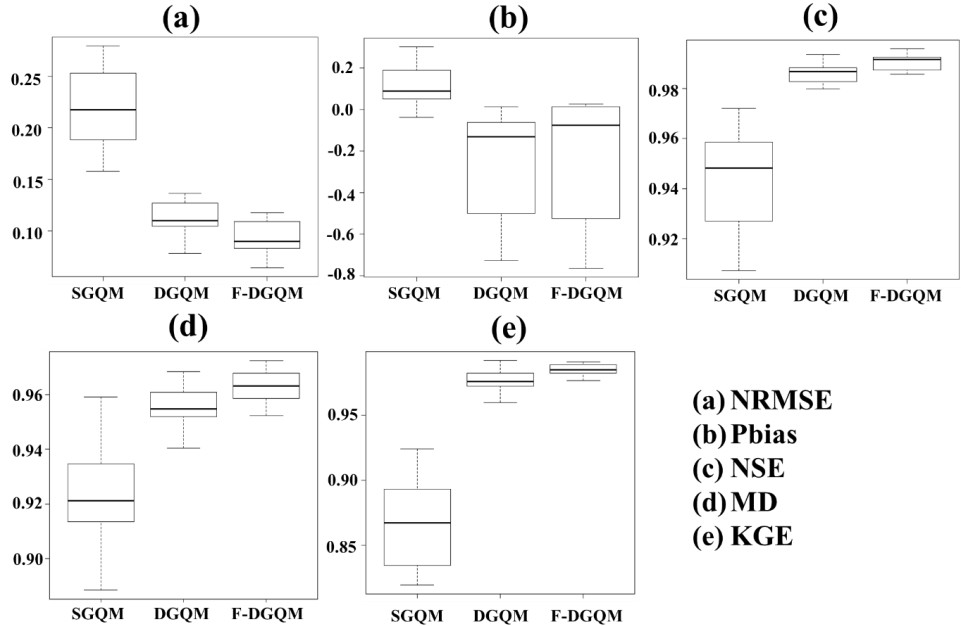

Figure 6. Performance of three QM methods in correcting GCM simulated monthly
precipitation bais at 22 stations in South Korea.

Figure 7 shows the performance of PDFs and CDFs of bias-corrected precipitations of SGQM,
DGQM, and F-DGQM at 22 stations based on KLD and JSD. Overall, the PDFs and CDFs of
F-DGQM were most similar to the observation than the other two methods. In contrast, the
PDFs and CDFs of SGQM showed the largest difference from the observation. The results
indicate the better reproducibility of observed precipitation PDF and CDF using F-DGQM.



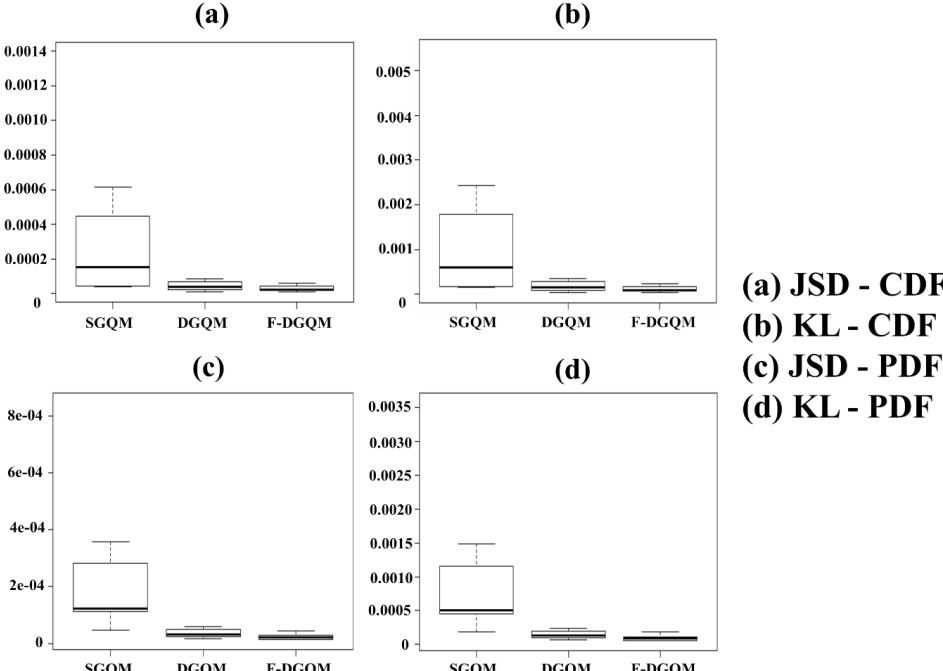

(a) JSD - CDF
(b) KL - CDF
(c) JSD - PDF
(d) KL - PDF


Figure 7. Comparison of bias-corrected monthly precipitation of 8 CMIP6 GCMs at 22
stations using three QM methods based on Kullback–Leibler and Jensen-Shannon
divergence.

The scatter plots of the bias-corrected monthly precipitation using the methods against
observations are shown in Figure 8. Overall, the scatter plots showed a remarkable
improvement in F-DGQM bias-corrected precipitation in association with observation. The
SGQM tended to inflate or underestimate observation significantly. Although the difference
between F-DGQM and DGQM was not high, the F-DGQM showed a more improvement in
precipitation reproducibility.



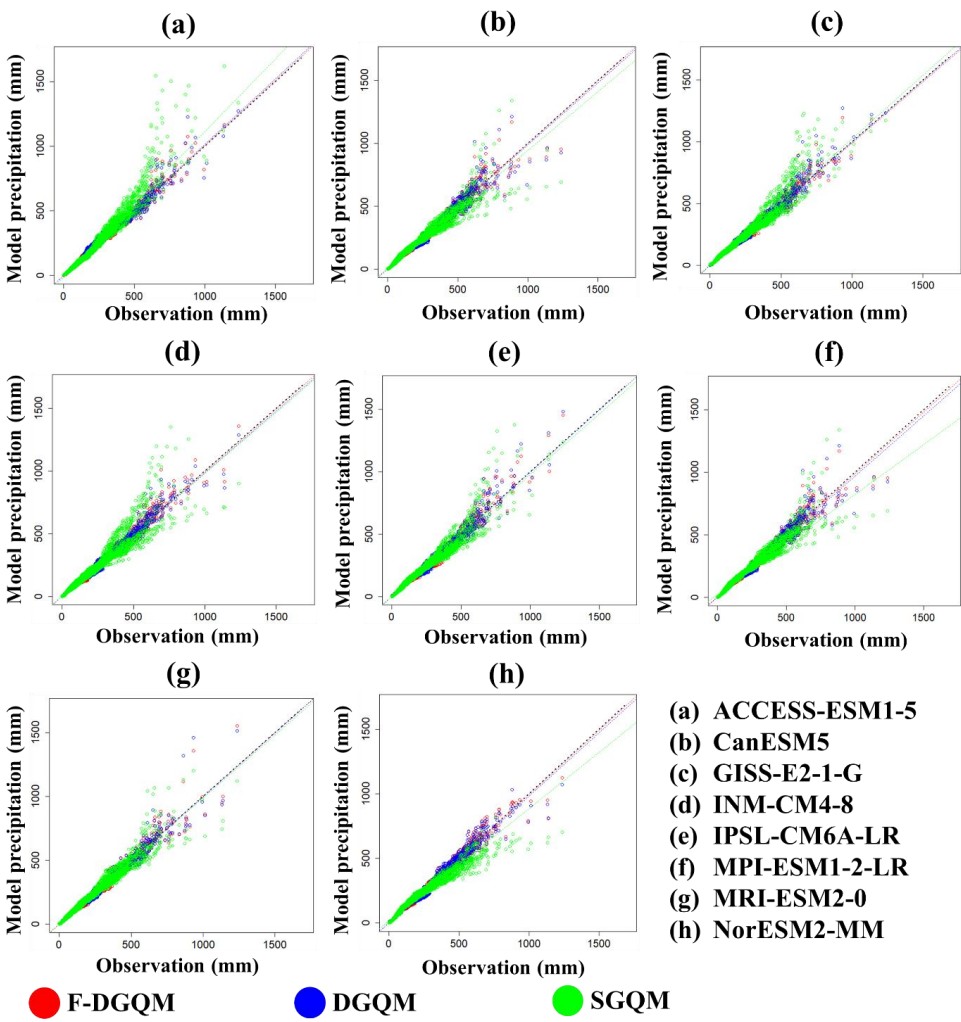

Figure 8. Performance comparison of three quantile mapping methods in correcting bias in 8
CMIP6 GCMs at 22 stations based on scatter plot

**4.1.3 Comparison at each station**

Figure 9 presents the average RMSE in bias-corrected precipitation of 8 GCMs at 22 stations
using different QM methods. The figure shows that the performance of F-DGQM was higher
than the other two methods at all stations. DGQM showed a better performance than SGQM
but lower than F-DGQM at all locations.



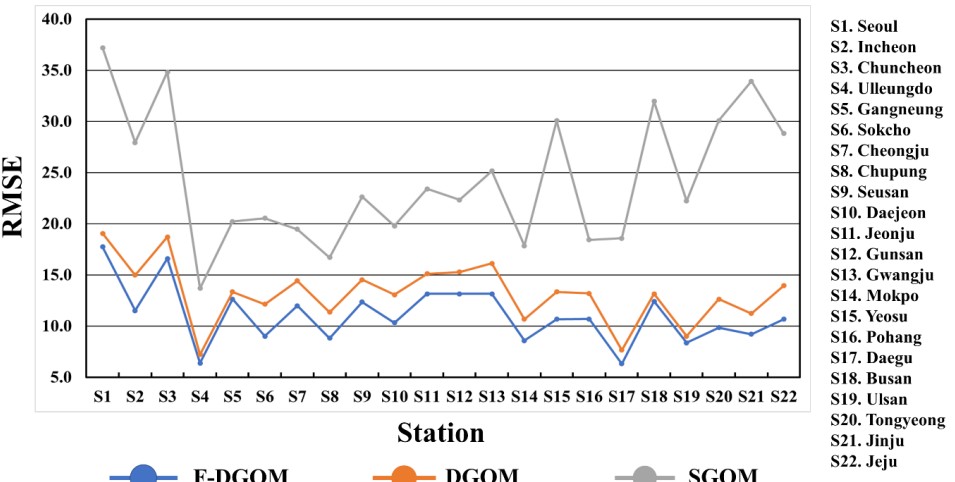

S1. Seoul
S2. Incheon
S3. Chuncheon
S4. Ulleungdo
S5. Gangneung
S6. Sokcho
S7. Cheongju
S8. Chupung
S9. Seusan
S10. Daejeon
S11. Jeonju
S12. Gunsan
S13. Gwangju
S14. Mokpo
S15. Yeosu
S16. Pohang
S17. Daegu
S18. Busan
S19. Ulsan
S20. Tongyeong
S21. Jinju
S22. Jeju

Figure 9. The RMSE in bias-corrected monthly precipitation of 8 CMIP6 GCMs using three quantile mapping methods at 22 stations.

This study also compared the performance improvement using other five statistical metrics listed in the method section and presented in Table S2. Overall, the performance of F-DGQM was higher than DGQM and SGQM in all metrics. F-DGQM showed a higher improvement in bias-corrected precipitation at Jinju, where the precipitation is relatively low (Average improvement 55%). Furthermore, the average improvement using F-DGQM compared to DGQM in all stations was 16%. The results indicate that a flexible quantile division location significantly improves the bias correction performance.

**4.2 Flexible double distribution quantile mapping**

**4.2.1 Results of $\delta$ and distribution selection**

The performance of the QM method by selecting the appropriate distribution fitted on two parts divided based on optimum $\delta$ is presented in this section. The best distributions determined for above and below of the selected $\delta$ at 22 stations are provided in Table S3. Overall, the Weibull exhibited the best performance for below $\delta$ for GCMs and observed precipitation (110 times), followed by Gamma (61 times) and Lognormal (5 times). The Weibull was also the best in fitting GCMs and observed data above $\delta$ (112 times), followed by Gamma (59 times) and Lognormal (5 times).



Table S4 presents the δ of F-DDQM based on the RMSE results at 22 stations. Figure 10
presents the number of δ selected at 22 stations based on the RMSE using a heatmap. The most
selected δ for CMIP6 GCMs was 80[th] and 95[th] quantiles. However, most GCMs had closer to
optimum RMSE for higher quantiles (88%-95%) than the lower quantiles (80%-87%). For
example, the most δ of GISS-E2-1-G, INM-CM4-8, IPSL-CM6A-LR, and MRI-ESM2-0 was
95[th] quantile. On the other hand, the most δ of ACCESS-ESM1-5, CanESM5, MPI-ESM1-2-
LR, and NorESM2-MM was 80[th] quantile. Therefore, the 80[th] and 95[th] quantiles were the best
δ for the GCMs, whereas the 89[th] quantile was never chosen.

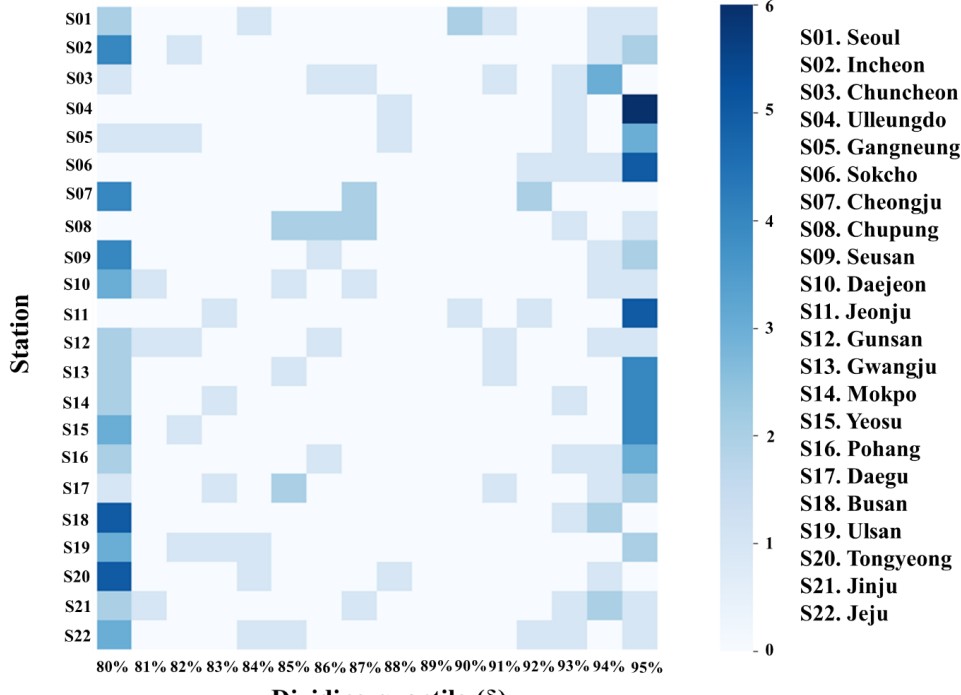


Figure 10. The heatmap shows the number of selected δ depending on RMSE

**4.2.2 Evaluation results for double distribution quantile mapping**
The precipitation of 8 CMIP6 GCMs was bias-corrected at 22 stations using F-DDQM with
selected δ and distributions.
The performance of the bias-corrected precipitation using F-DDQM, F-DGQM and DGQM at
22 stations based on five evaluation metrics is presented in Figure 11. The results showed that
the median NRMSE of bias-corrected precipitation using F-DDQM was higher than the other



two methods. The Pbias showed that DGQM and F-DGQM underestimated, whereas the F-
DDQM overestimated the monthly precipitation. The median Pbias of F-DDQM and F-DGQM
was closer to the optimal value. The median NSE of F-DDQM was closer to the optimal value
than that for F-DGQM and DGQM. In addition, the median MD of F-DDQM was the highest.
The median KGE of F-DDQM was also slightly higher than F-DGQM.

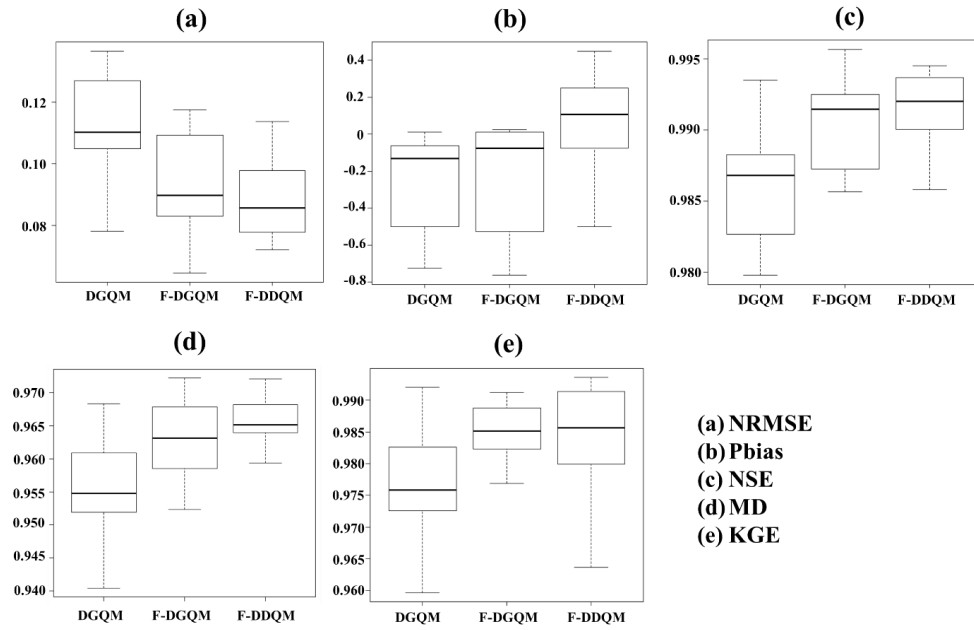


Figure 11. The performance of DGQM, F-DGQM, and F-DDQM in correcting GCM
simulated monthly precipitation bais at 22 stations based on five statistical metrics.

The performance of the methods based on JSD and KLD is shown in Figure 12. Both metrics
showed that PDF and CDF of F-DDQM corrected precipitation were closer to the observation.
F-DGQM performed better than DGQM but much lower than F-DDQM.

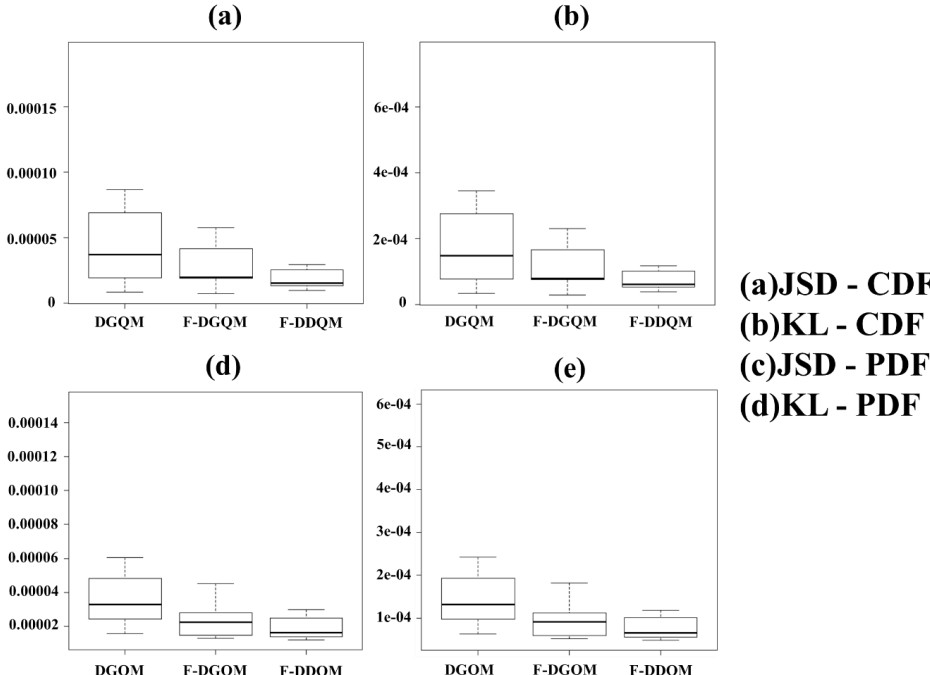

(a)JSD - CDF
(b)KL - CDF
(c)JSD - PDF
(d)KL - PDF

Figure 12. Comparison of DGQM, F-DGQM, and F-DDQM methods in bias correcting monthly precipitation of 8 CMIP6 GCMs at 22 stations using Kullback–Leibler and Jensen-Shannon divergence.

**4.2.3 Comparison of performance at each station**

Figure 13 presents the average RMSE in bias-corrected precipitation of 8 CMIP6 GCMs at 22 stations. The figure shows lower RMSE for F-DDQM at all stations than the other two methods. The performance of the methods based on other statistical metrics is presented in Table S6. The results showed average improvement using F-DDQM was 1.1% and 3.3% in RMSE compared to F-DGQM and DGQM. These results indicate an improvement in precipitation bias correction using F-DDQM at different locations having diverse climates.



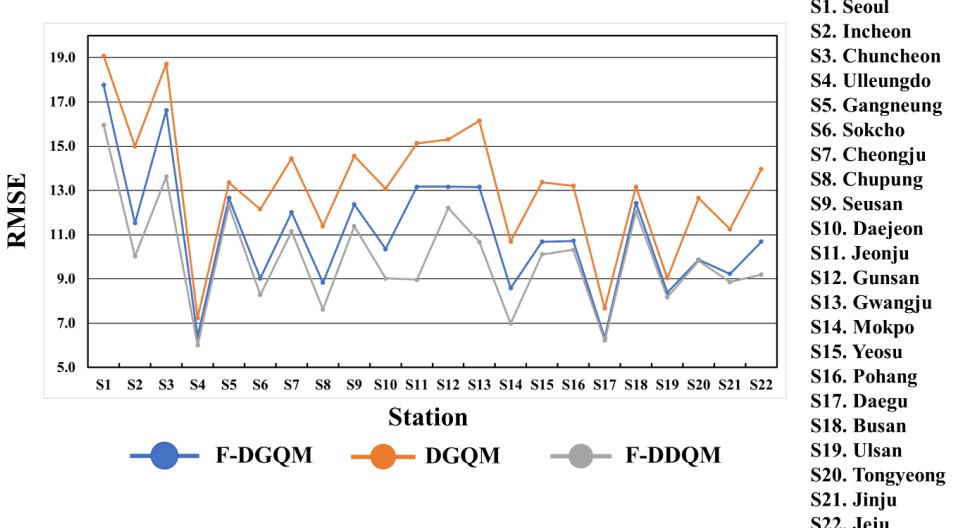

S1. Seoul
S2. Incheon
S3. Chuncheon
S4. Ulleungdo
S5. Gangneung
S6. Sokcho
S7. Cheongju
S8. Chupung
S9. Seusan
S10. Daejeon
S11. Jeonju
S12. Gunsan
S13. Gwangju
S14. Mokpo
S15. Yeosu
S16. Pohang
S17. Daegu
S18. Busan
S19. Ulsan
S20. Tongyeong
S21. Jinju
S22. Jeju


Figure 13. The RMSE in bias-corrected monthly precipitation of 8 CMIP6 GCMs at the 22
stations using F-DGQM, F-DDQM, and DGQM.

Figure 14 shows the relative performance of F-DGQM and F-DDQM in correcting
precipitation using scatter plots. Overall, F-DDQM improved precipitation performance than
F-DGQM. The correlation between F-DDQM corrected and observed precipitation was
slightly higher than that obtained for F-DGQM for all GCMs.

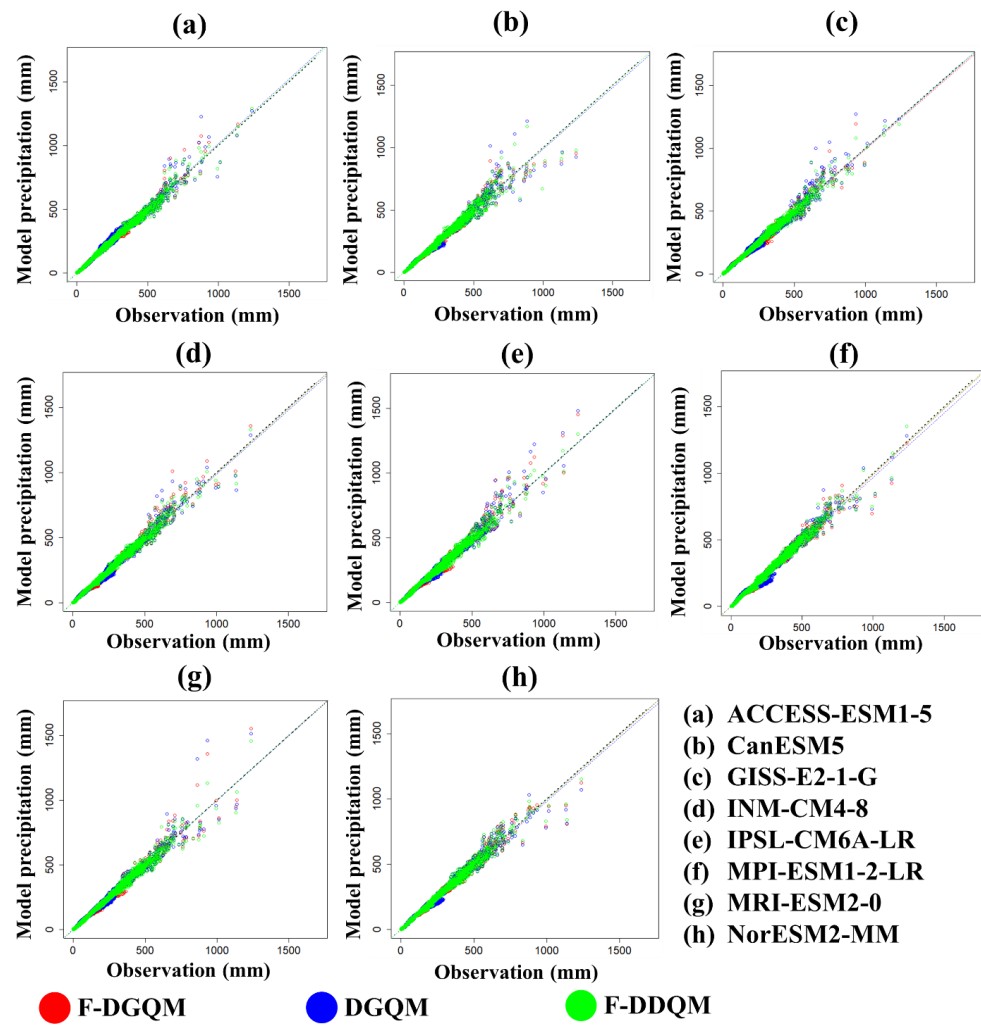

Figure 14. Performance comparison of F-DGQM, F-DDQM, and DGQM methods in
correcting bias in 8 CMIP6 GCMs at 22 stations based on scatter plot

## 4.3 Performance comparison based spatial precipitation indices

The performance of bais corrected precipitation using F-DGQM, F-DDQM, DGQM and
SGQM in simulating the spatial distribution of observed maximum precipitation, median
precipitation and standard deviation of precipitation are presented in Figure 15 (a), (b), and (c),
respectively. Overall, the spatial distribution of maximum precipitation estimated using F-
DDQM was closer to the observation. The maximum precipitation obtained using SGQM





tended to inflate in the northwest, where extreme precipitation occurs more, whereas it
underestimated maximum precipitation in the south. The error in DGQM maximum
precipitation was narrower than SGQM, but it overestimated maximum precipitation in some
regions. F-DGQM captured maximum precipitation in the central region similar to DGQM. In
contrast, F-DDQM showed the smallest difference with the observation in most regions and
the highest performance.

The precipitation median estimated by SGQM was higher than the observation in the

western region. DGQM estimated a smaller precipitation median than SGQM in most regions,
whereas overestimated it in the southwest region. F-DGQM showed a negligible difference
with observed median precipitation (less than 5 mm in most regions). However, the smallest
difference with the observed median precipitation was obtained using F-DDQM.

The difference in precipitation standard deviation between SGQM corrected and

observed precipitation was the largest (above 5 mm in most regions) compared to other
methods. The DGQM showed a smaller difference than SGQM, but it overestimated the
standard deviation in some regions. In contrast, F-DGQM showed the lowest difference with
observed precipitation standard deviation in most regions (the difference was close to zero).
These results indicated better performance of F-DGQM and F-DDQM in caption spatial
distribution of precipitation indices. However,    F-DDQM showed slightly better performance
than F-DGQM.



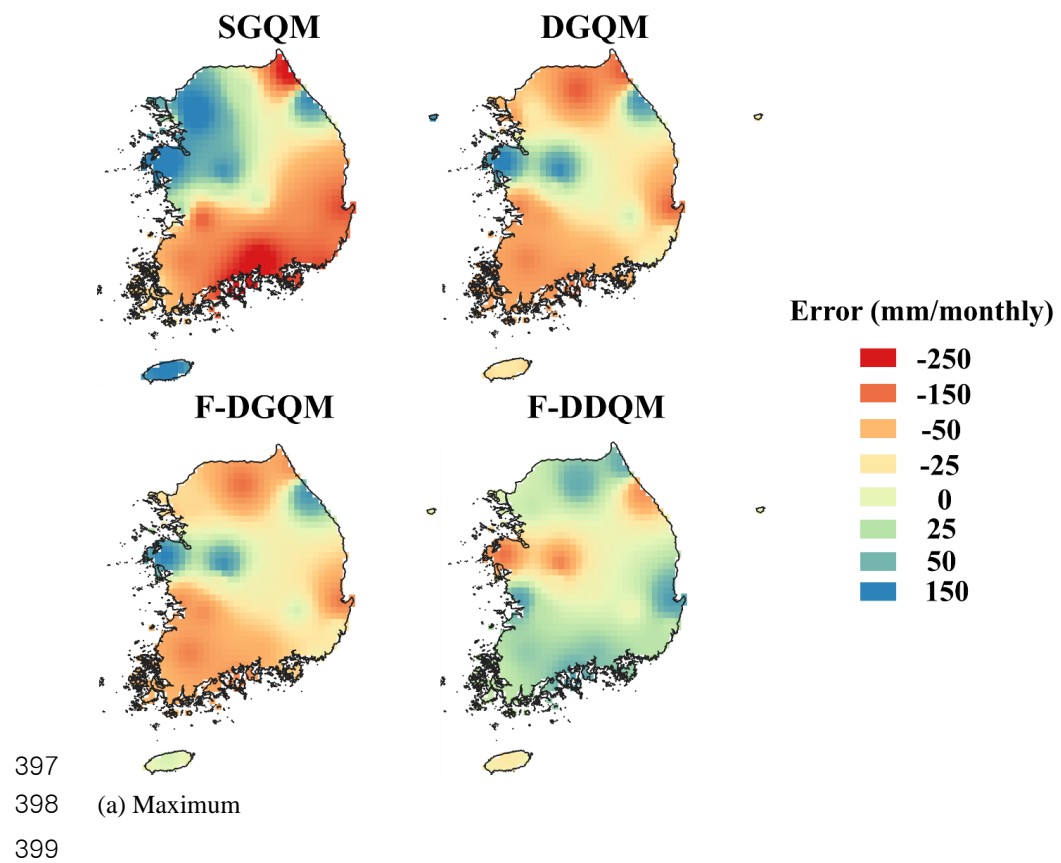


(a) Maximum



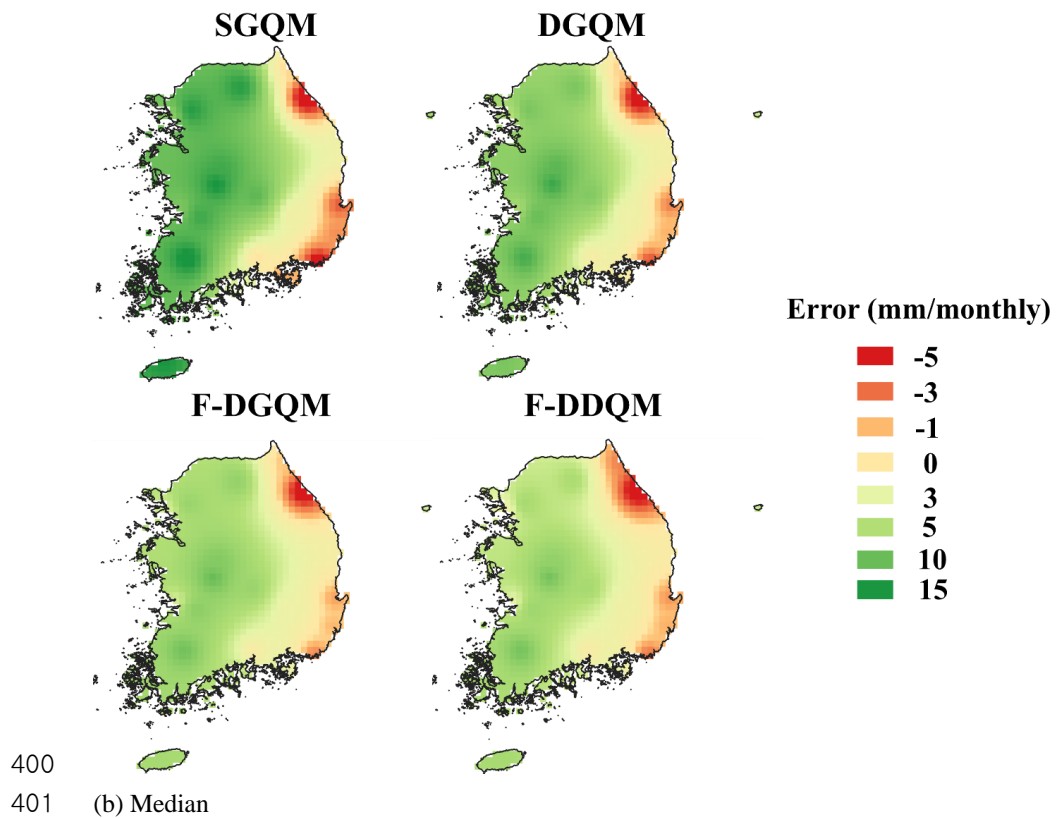


(b) Median



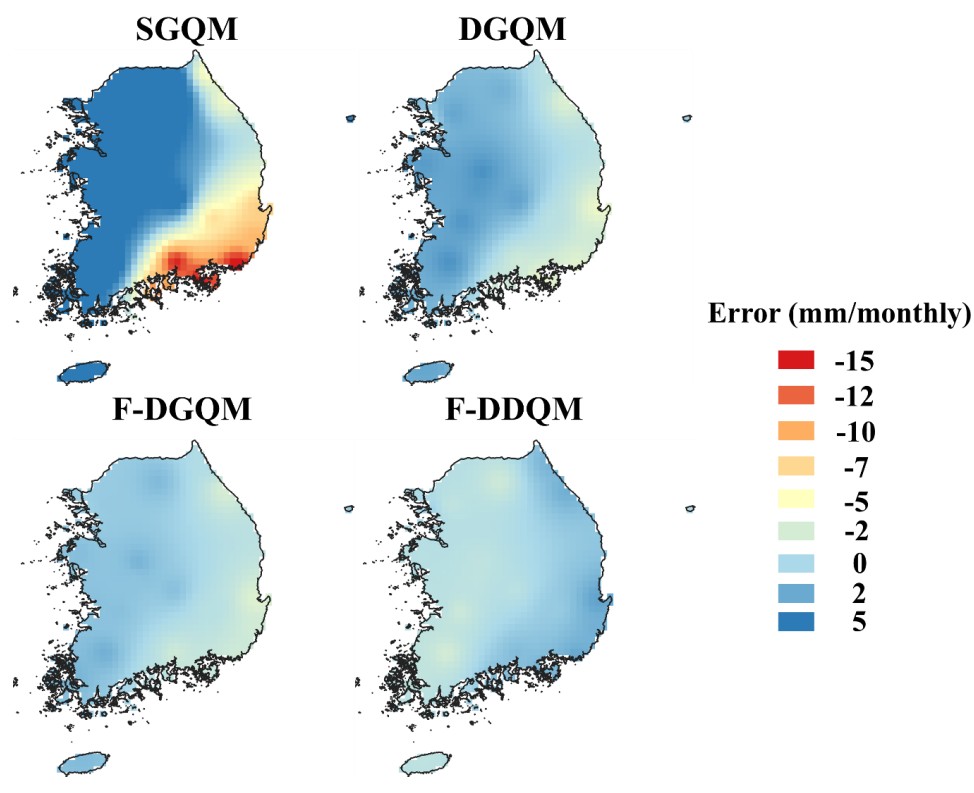


(c) Standard Deviation
Figure 15. Performance of different bias correction methods in reconstructing the spatial
distribution of three precipitation metrics: (a) maximum precipitation; (b) median
precipitation; and (c) standard deviation of precipitation for the base period 1970-2014

The average error in estimating the spatial distribution of three precipitation metrics by the bias
correction methods is presented in Table 2. Overall, F-DDQM showed the lowest variance with
respect to observation in all metrics. The difference between the F-DDQM corrected and
observations maximum precipitation was 50.6 mm, median precipitation was 4.5 mm, and
standard deviation was 1.2, which were the lowest among all methods.

Table 2. Errors (mm) in estimating observed precipitation metrics using different bias
correction methods.

| Metrics | SGQM | DGQM | F-DGQM | F-DDQM |
|---------|------|------|--------|--------|





| | | | | |
|---|---|---|---|---|
| SD | 10.7 | 2.2 | 1.3 | 1.2 |
| Max | 151.2 | 82.7 | 70.2 | 50.6 |
| Median | 8.2 | 6.1 | 4.9 | 4.5 |


**4.4 Generalized extreme value of the bias corrected precipitation**
This study compared the extreme values of historical bias-corrected precipitation four QM
methods based on GEV distribution. The precipitation for above the 95th percentiles are
presented in Figure 16. L-moment was used to estimate the GEV parameters of bias-corrected
GCMs using four QM methods. Overall, the PDF of F-DDQM was the most similar to the
observed PDF. Although the extreme precipitation of MRI-ESM2-0 was slightly higher than
the observed, most GCMs showed similar extreme precipitation to the observed. F-DGQM
estimated extreme precipitation was closer to observed than DGQM and SGQM, but its
performance was lower than F-DDQM. The results indicate F-DDQM is the best in correcting
bias in precipitation extremes.

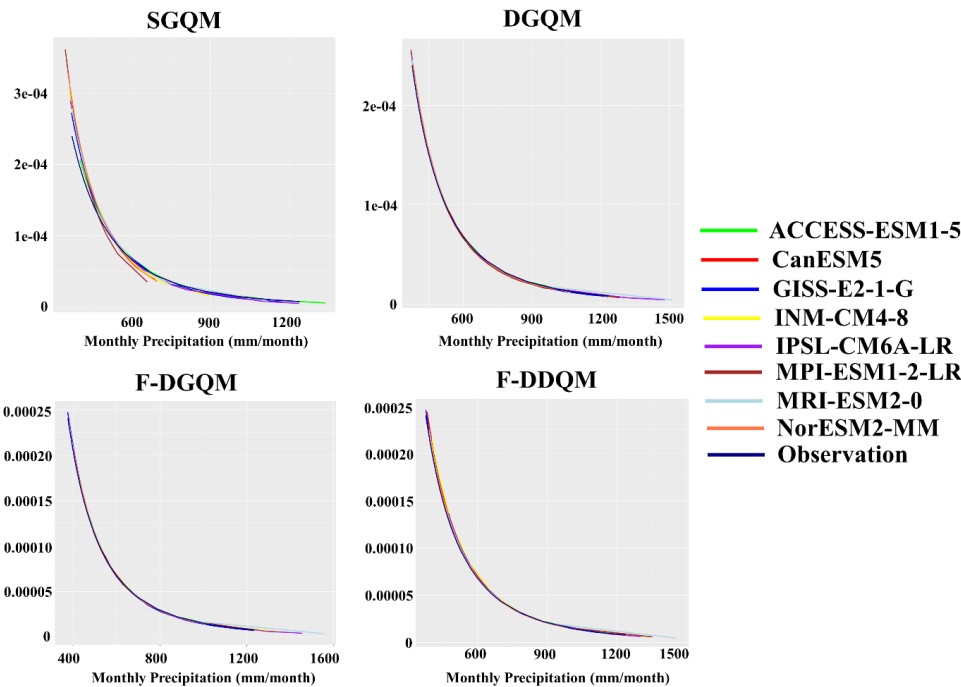


Figure 16. Comparison of PDF above the 95th percentile based on GEV distribution at 22
stations estimated using four QM methods.






The differences between the observed and bias-corrected precipitation GEV distributions were
estimated using KLD and JSD. The obtained results for all the GCMs are presented using
boxplots Figure 17. Overall, the GEV distribution of the F-DDQM for both divergences was
the closest to the observed in PDF and CDF, followed by F-DGQM, DGQM and SGQM. The
again proves the capability of F-DDQM in correcting bias in precipitation extremes.

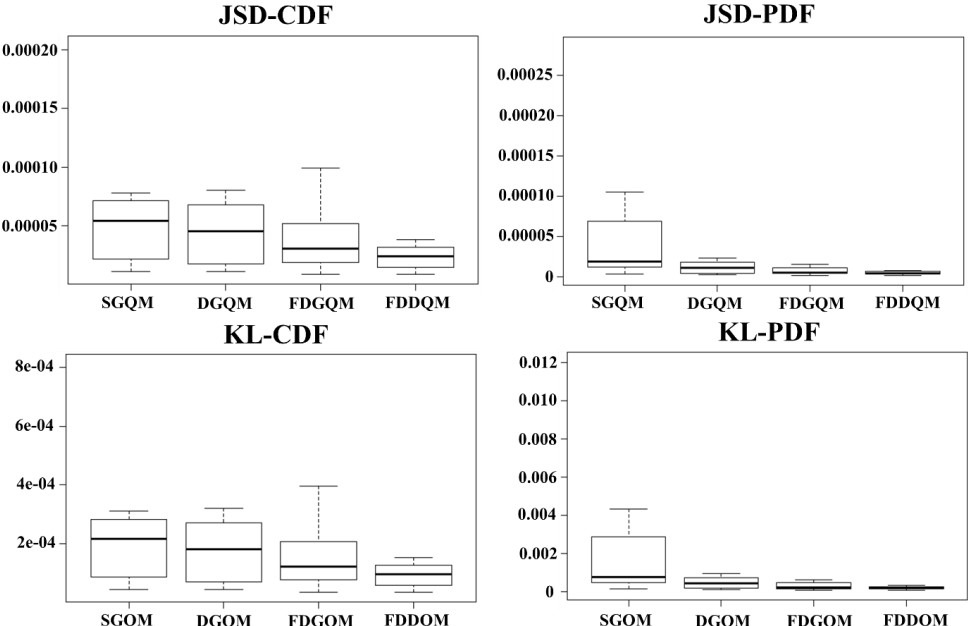


Figure 17. Differences in GEV distribution between the observed and bias-corrected GCMs'
precipitation at 22 stations using KLD and JSD.

.



## 5. Discussion

Although the QM algorithms can effectively eliminate biases and errors in GCM simulations, the performance is dependent on the QM method, such as non-parametric transformation, parametric transformation, and distribution derived transformations (Song et al., 2020). The distribution-derived transformation was developed by combining distribution functions like Bernoulli-Gamma. Various functions have been applied to improve its performance (Gudmundsson et al., 2012; Cannon et al., 2015). Nevertheless, the general QM can artificially impair trends in future projections (Cannon et al., 2015). Therefore, improving the GCM's extreme precipitation bias correction method is important. DGQM was the proposed method to solve this problem. However, there is no clear reason for determining 90% or 95% (Pastén-Zapata et al., 2020; Yang et al., 2010) as the dividing point. Furthermore, the gamma probability function is generally used to fit two divided segments, but it is not the most appropriate probability distribution function at all locations. Therefore, this study presented F-DGQM and F-DDQM that determines $\delta$ according to optimum RMSE, considering two independent probability distributions for two divided segments.

The $\delta$ of F-DGQM was the 80th quantile in this study based on the RMSE at most stations. Conversely, the second-highest performing $\delta$ was the 95th quantile. It means that the suitable $\delta$ is different at different stations depending on the scale and shape of the GCM precipitation distribution. Therefore, the determination of $\delta$ can affect the difference between extreme and mean precipitation. Therefore, it was reasonable to use RMSE to determine double distribution.

The bias correction performance of F-DGQM showed a large improvement, as shown in Figure 6, in all evaluation metrics compared to DGQM and SGQM. The PDFs and CDFs of the bias-corrected precipitation using three QM methods were compared with the observed PDF and CDF using JSD and KLD. F-DGQM showed better performances than DGQM and SGQM. However, only $\delta$ determination does not guarantee the superior performance of F-DGQM than other methods. The Gamma distribution may not show the best performance at all stations and all GCMs. The combination of different distributions proposed by Gudmundsson et al. (2012) can improve the bias correction performance over a single distribution. Therefore, this study proposed F-DDQM, considering suitable distributions for two individual segments.

The performance of F-DDQM showed better performance than F-DGWM because of considering three probability distributions for two individual segments. The most selected $\delta$ in





F-DDQM showed that the high percentiles (88%-95%) were selected more than the low
percentiles (80%-87%). Therefore, it can be remarked that a suitable δ can be selected at a
relatively high quantile. Furthermore, the Weibull distribution performed best for below δ.
Furthermore, Weibull performed best for above δ, followed by Gamma. These results prove
that the Lognormal PDF is not proper in analyzing the monthly precipitation of South Korea.
The performance of F-DDQM was higher than F-DGQM in all evaluation metrics. Furthermore,
the performance improvement using F-DDQM was more than F-DGQM at all stations.
This study also presented the spatial differences between the observed and the bias-
corrected monthly precipitation metrics (Figure 15). Overall, the performance of F-DDQM was
the highest. The F-DDQM estimated spatial distribution of all three metrics very similar to
observation at all regions of South Korea. On the other hand, SGQM overestimated the
maximum precipitation, and thus, the corrected precipitation tends to be inflated for the most
frequent values (Cannon et al. al., 2015; Teng et al., 2015; Yang et al., 2010). The results clearly
showed that the F-DGQM and F-DDQM improved the performance of the existing versions of
QM bias correction methods. The performance of F-DDQM is the best among all. Furthermore,
uncertainty in F-DDQM corrected bias is relatively low.
The GEV distribution of F-DDGM precipitation was also more similar to the observed
precipitation compared to the others. The JSD and KLD also showed that F-DDGM corrected
precipitation PDF and CDF are closest to the observed PDF and CDF at all stations. The results
indicate the higher performance of F-DDQM in various aspects.
**6. Conclusions**
In this study, two new bias correction methods were proposed to improve the performance of
double gamma quantile mapping, F-DGQM and F-DDQM. F-DGQM determines δ based on
RMSE to distinguish two segments of the gamma distribution for bias correction. F-DDQM
uses the optimal probability distribution for two segments defined by δ to improve bias
correction. Furthermore, the performance of F-DGQM and F-DDQM, proposed in this study,
was compared with two existing QM methods, DGQM and SGQM, which have been widely
used in different regions for correcting bias in monthly precipitation. This study concluded the
following: First, the performance of F-DGQM is generally higher than SGQM and DGQM at
all stations. Second, the δ (dividing point) of F-DGQM and F-DDQM varies from station to
station, indicating a constant δ at all stations is not optimal for bias correction. Third, the



judicious selection of the dividing point improves the performance for bias correction. Fourth, F-DDQM corrected precipitation has lower uncertainty than other methods. Fifth, F-DDQM performs best in correcting bias in extreme precipitation.

This study contributes to technological development by suggesting a new bias correction method that can be used more flexibly than the existing DGQM for reliable correction of GCM biases.

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
