# Peer review of "Development of flexible double distribution quantile mapping for better"

_Hydrology and Earth System Sciences, 2022_

## Author Comment (AC2)

**Reviewer 2**

**Comment**

The section of Introduction is not well written, and the logic is not so clear. For example, what is the relationship between the paragraph 2 and 3? In Paragraph 2, you introduced the advantage of QDM method that can address the drawbacks of QM and also cited several different categories of methods developed based on QDM, thus according to the normal logic, it should describe the method of QDM in Paragraph 3 rather than the QM. In addition, without giving the reasons that "QM does not always outperform other bias-correction methods at all locations", how did you get the conclusion of "this emphasizes choosing an appropriate probability distribution function for successful bias correction"? Furthermore, the objectives/problems aimed to be achieved/addressed are not properly stated in the last paragraph.

**Answer**

Thanks for your comment. Paragraph 2 of the introduction describes previous studies to solve problems caused by QM methods. Therefore, we added the following sentences to improve the readability of the text [L14-15]:

The distribution-derived transformations, such as quantile mapping (QM), are most widely used for bias corrections because of their simplicity, ease of access, and higher proficiency (Ringard et al., 2017; Maraun et al., 2010; Ines and Hansen, 2006; Li et al., 2010). However, the QM shows high performance in bias correcting stationary climate variables but low reliability for nonstationary data. This universal problem has prompted several studies to improve the performance of QM techniques.

Also, we have revised the sentence in paragraph 3 of the introduction to read:
Nevertheless, QM does not always outperform other bias-correction methods at all locations (Song et al., 2020). This emphasizes choosing an appropriate distribution function for successful bias correction.

In addition, to secure the objectives of this study, we have modified it as follows:

This study aimed to propose a new flexible double distribution quantile mapping (F-DDQM) method considering adjustable dividing points and two individually selected distributions for two segments to improve the performance of DGQM. Three PDFs, Weibull, lognormal, and

Gamma distributions, were considered for selecting appropriate PDFs for two segments. The dividing point was determined based on the normalized five-evaluation metrics of the overall precipitation distribution. The proposed method was employed to correct the bias of 8 GCMs of Coupled Model Intercomparison Project 6 (CMIP6) at 22 stations in South Korea. The performance of the proposed method was compared with the DGQM and the Flexible DGQM (F-DGQM) using five evaluation metrics. Furthermore, the performance of the proposed method in correcting the bias of extreme precipitation was compared to previous methods based on GEV distribution. Besides, the difference between the simulated precipitation distribution and the observed distribution was compared using Jensen-Shannon (JSD) and Kullback-Leibler divergence (KLD). This study contributes to improving the bias correction method for the better projection of extremes.

**Comment**

In this study, using the monthly-scale precipitation to test the effectiveness of the newly proposed flexible double distribution quantile mapping method in correcting the bias of GCM is not proper and it cannot well reflect the extreme precipitation characteristics. The authors must valid the performance of this method in bias-correcting of daily-scale GCMs, the daily data present larger spatial variability and are more useful for climate change studies.

**Answer**

Thanks for your comment. Based on the comments of most reviewers, we changed the proposed bias correction method based on the daily precipitation.

**Comment**

This paper is written casually, and there exist many grammar and tense problems, which needs to be polished by native English speakers.

**Answer**

Thanks for your comment. In response to your comments, we have solved all the grammar and tense problems in the text

**Comment**

Line 14-15: It is not appropriate to directly use the 90th quantile to reflect the question, because in many references they may also using the 95th or 99th quantile rather than only the 90th quantile.

**Answer**

Thanks for your comment. In response to your comments, we modified the following sentence [L14-15]:

The double gamma quantile mapping (DGQM) can outperform single gamma quantile mapping (SGQM) for bias correction of global circulation models (GCMs) using two gamma functions for two segments based on a specific quantile.

**Comment**

Line 15: "Gamma probability distribution function" instead of "Gamma probability function".

**Answer**

Thanks for your comment. In response to your comments, we modified the following sentence [L15]:

However, there are two ambiguous points: the specific quantile and considering only the Gamma probability distribution function.

**Comment**

Line 17: "consider" instead of "considered".

**Answer**

Thanks for your comment. In response to your comments, we modified the following words:

Therefore, this study introduced a flexible dividing point, $\delta$ (%), which can be adjusted to the regionally observed values at the station and consider the combination of various probability

distributions for the two separate segments (e.g., Weibull, lognormal, and Gamma).

**Comment**

Line 18: add "e.g." before "Weibull, lognormal…..", add "the" before " two separate segments".

**Answer**

Thanks for your comment. In response to your comments, we modified the following words:

Therefore, this study introduced a flexible dividing point, δ (%), which can be adjusted to the regionally observed values at the station and consider the combination of various probability distributions for the two separate segments (e.g., Weibull, lognormal, and Gamma).

**Comment**

Line 20: delete "to correct bias".

**Answer**

Thanks for your comment. In response to your comments, we modified the sentence as following:

The newly proposed method, flexible double distribution quantile mapping (F-DDQM), was employed to correct the bias of 8 GCMs of Coupled Model Intercomparison Project Phase 6 (CMIP6) at 22 stations in South Korea.

**Comment**

Line 21 and 23: the tense is wrong. "show" instead of "show".

**Answer**

Thanks for your comment. In response to your comments, we modified the word as following:

The results clearly show a higher performance of F-DDQM than DGQM and Flexible-DGQM

(F-DGQM) by 27% and 19%, respectively, in root mean square error.

**Comment**

Line 25: delete "the" before "better projection of extreme values".

**Answer**

Thanks for your comment. In response to your comments, we modified the word as following:

This study contributes to improving the bias correction method for better projection of extreme values.

**Comment**

Line 45: why use "but" when there is no turning point?

**Answer**

Thanks for your comment. In response to your comments, we modified the word as following:

The distribution-derived transformations, such as quantile mapping (QM), are most widely used for bias corrections because of their simplicity, ease of access, and higher proficiency (Ringard et al., 2017; Maraun et al., 2010; Ines and Hansen, 2006; Li et al., 2010).

**Comment**

Line 55: "and" <- "with" or "using".

**Answer**

Thanks for your comment. In response to your comments we modified the word as following:

The QM method replaces the quantiles of simulated data corresponding to a given probability with the observed quantile corresponding to the same probability (Cannon, 2008; Piani et al., 2010; Cannon, 2012; Heo et al., 2019).

**Comment**

Line 75: "is aimed to propose" instead of "proposed".

**Answer**

Thanks for your comment. In response to your comments, we modified the word as following:

This study aimed to propose a new flexible double distribution quantile mapping (F-DDQM) method considering adjustable dividing points and two individually selected distributions for two segments to improve the performance of DGQM. Three PDFs, Weibull, lognormal, and Gamma distributions, were considered for selecting appropriate PDFs for two segments.

**Comment**

Line 77: Why do you choose these three PDFs since you mentioned that the most appropriate distribution can be different for different regions in Line 72? Whether the only three PDFs are too few? Are these three PDFs suitable for precipitation extremes, i.e., the segment larger than the given threshold like 90th quantile? Are these three PDFs suitable for precipitation extremes, i.e., the segment larger than the given threshold like 90th quantile?

**Answer**

Thanks for your comment. The three distributions used in this study are objective functions commonly used to apply the quantile mapping method. Therefore, we compare to apply the three distributions at 22 stations. In futher studies, we will apply various objective functions. Regarding the limitations of this study, we added the following sentence to section 6:

The present study considered only three distribution functions and five evaluation metrics. An attempt can be taken in the future to improve the calibration performance by adding evaluation metrics and distribution functions.

**Comment**

Line 79: Why only use RMSE to select the dividing point? Are there any indicators that are more suitable to select the dividing point?

**Answer**

Thanks for your comment. We selected dividing points using five evaluation metrics to improve the F-DGQM and F-DDQM of this study. Normalization was applied to the results of the five evaluation metrics, and the quantile with the highest average value was used for 22

stations.

**Comment**

Line 81: add "method" after "The performance of the proposed".

**Answer**

Thanks for your comment. In response to your comments, we modified the word as following:

The performance of the proposed method was compared with the DGQM and the Flexible DGQM (F-DGQM) using five evaluation metrics.

**Comment**

Line 83-84: What do you mean "the performance …………based on GEV distribution? It doesn't seem to be a complete sentence.

**Answer**

Thanks for your comment. We complete the sentence as follows:

Furthermore, the performance of the proposed method in correcting the bias of extreme precipitation was compared to previous methods based on GEV distribution.

**Comment**

Line 126: "based on" instead of "for".

**Answer**

Thanks for your comment. In response to your comments, we have modified sentence as follows:

Equation 2 computes the interpolation weight based on the distance between the grid and the interpolation points.

**Comment**

Line 127-134: What's the meaning of those variables in Eq. (1) – Eq. (2)? They are not stated

properly and could not be well understood. How do you determine the surrounding grids close to one specific location that are used in Eq. (1) – Eq. (2)?

**Answer**

Thanks for your comment. We describe the corresponding variables in Eq 1 and Eq 2 as follows. Eq 1 is to estimate the interpolated precipitation, and Eq 2 accounts for the interpolation weights in Eq 1. Here, the description of $D_{(x,x_s)}^c$ is presented in the text. Furthermore, interpolated grids in this study used grids close to each station.

For example, if we interpolate the precipitation for the Seoul station (Lat: 37.57, Lon: 126.96), we used 50 adjacent grids to calculate the interpolated precipitation. (e.g. grid1: 37.125, 126.375; grid2: 37.375, 126.375; grid3: 37.125, 126.625 ...... )

**Comment**

Line 144: Is the F-1g is Eq. (3) correct? It is very easy to consider F-1g as the converse function of Fg. Please explain the variables in a proper way.

**Answer**

Thanks for your comment. In response to your comments, we have modified Equation 3 as follows:

$$P_g(t) = F_o^{-1}\big(F_g(P_m(t), \alpha_m, \beta_m), \alpha_o, \beta_o\big) \qquad (3)$$

The modified the paragraph as follows:

where $P_g(t)$ denotes the bias-corrected monthly precipitation, $P_m(t)$ represents GCM raw data, $F_o^{-1}$ is the inverse CDF of the observed data to which the gamma function is applied, and $F_g$ is the CDF of the GCM outputs. $\alpha_o$, $\alpha_m$, $\beta_o$ and $\beta_m$ represent shape and scale parameters of observed and GCM simulation, respectively.

**Comment**

Line 179: delete "the" before "other climate variables".

**Answer**

Thanks for your comment. In response to your comments, we modified paragraph as follows:

The proposed method can be used for bias correction of various climate variables. However, since the natural variability of precipitation is higher than other climate variables, this study considered only precipitation bias correction (Deser et al., 2012; Cannon et al., 2015).

**Comment**

In Section 3.3-3.4: These two parts are the core contents of this study, but the relevant information is two little. The detailed calculation process should be described here.

**Answer**

Thanks for your comment. In response to your comments, we have modified Sections 3-3 and 3-4 as follows L164-L217

**Comment**

In Line 160: Why do you choose the δ values between 80%-95%? It seems also very random like other studies. In addition, how do you use RMSE to determine the δ has not been clearly given.

**Answer**

Thanks for your comment. To respond to your comments, we've considered the range of the quantiles as much as possible before the no-precipitation interval. Therefore, all stations perform bias correction starting from the quantiles that do not include days without precipitation. As a result, most stations will generally find an appropriate delta from 70% to 99% of the quantiles.

Further, we determine deltas based on five evaluation metrics that can reflect bias, error, and correlation coefficients.

**Comment**

Line 161: "determine" instead of "determined". Add "the" before "optimal RMSE".

**Answer**

Thanks for your comment. In response to your comments, we have modified paragraph as follows:

The upper δ is determined based on the optimal evaluation metrics of the distributions of 70–99% quantiles.

**Comment**

Line 169: Add "the" before "Gamma distribution". There are so many places where "the" has not been properly used or not been added. Please check it in the whole paper.

**Answer**

Thanks for your comment. In response to your comments, we have modified all text in this article.

**Comment**

In Figure 2 and 3, what are the differences? What are your points?

**Answer**

Thanks for your comment. Figure 2 performs the calibration using only the gamma function. Therefore, finding the appropriate delta at each station is the key. On the other hand, the critical point is to find the appropriate delta and function for the two segments, as shown in Figure 3.

**Comment**

Line 179: delete "the" before "other climate variables".

**Answer**

Thanks for your comment. In response to your comments, we have modified sentence as follows:

However, since the natural variability of precipitation is higher than other climate variables, this study considered only precipitation bias correction (Deser et al., 2012; Cannon et al., 2015).

**Comment**

Line 195: add "by" before "positive value".

**Answer**

Thanks for your comment. In response to your comments, we have modified sentence as follows:

Pbias represents the bias in the GCM and observation values. A positive Pbias indicates the tendency of overestimation and vice versa.

**Comment**

Line 202: Please check the correction of the Eq. (8).

**Answer**

Thanks for your comment. In response to your comments, we have modified Equation 8 as follows:

$$KGE = 1 - \sqrt{(r-1)^2 + (\alpha - 1)^2 + (\beta - 1)^2}$$

**Comment**

Line 208-219: In Section 3.6, the aim of using the Generalized extreme value distribution in this paper should be firstly explained.

**Answer**

Thanks for your comment. In response to your comments, we added a sentence in Section 3-6 as follows:

This study used a generalized extreme value (GEV) distribution to compare the extreme daily precipitation corrected by different bias correction methods.

**Comment**

Line 218: "bias-corrected precipitation" rather than "precipitation bias-corrected".

**Answer**

Thanks for your comment. In response to your comments, we have modified a sentence in Section 3-6 as follows:

This study compared the extreme values of daily bias-corrected precipitations using four QM methods considering GEV distribution.

**Comment**

Line 221: add the abbreviation KLD and JSD in the sub-title.

**Answer**

Thanks for your comment. This section has been included in sections 3-5 due to comments from other reviewers.

**Comment**

Line 245: The most selected quantile is the 80th that can be seen from Fig. 4, is this related with the lower bound of the $\delta$ values you set in this paper? This means whether the most selected $\delta$ value will be smaller than the 80th quantile if the lower bound of $\delta$ values is set lower than the 80th quantile. Similar for the second most selected 95th quantile.

**Answer**

Thanks for your comment. In response to your comments, we considered the maximum range excluding no rainfall, from 70% to 99%. Further, we found that the δ chosen was different for all stations. Therefore, the changed results are reflected in the text.

The modified Figure is as follows:

[Figure]

Figure 5. Heatmap showing the number of selected δ for F-DGQM depending on normalized values of five evaluation metrics at 22 stations

[Figure]

Figure 9. Heatmap showing the number of selected δ for F-DDQM depending on normalized five evaluation metrics at 22 stations

**Comment**

Line 245: The most selected quantile is the 80th that can be seen from Fig. 4, is this related with the lower bound of the δ values you set in this paper? This means whether the most selected δ value will be smaller than the 80th quantile if the lower bound of δ values is set lower than the 80th quantile. Similar for the second most selected 95th quantile.

**Answer**

Thanks for your comment. We also modified Figure 5 by changing to daily precipitation. Most δ were chosen from the middle quantiles. Therefore, we modified Section 4-3 based on the results.

**Comment**

Line 255-256: Please rewrite the title of Figure 5, same for Figure 10.

**Answer**

Thanks for your comment. In response to your comments, we have modified Figure 5.

**Comment**

Line 301: The tense should be the present tense when describing the founded results. Please check in the whole paper.

**Answer**

Thanks for your comment. We have modified the tense when describing the founded results in this paper.

**Comment**

Line 312: add "distribution" after "Weibull". Please check the tense.

**Answer**

Thanks for your comment. In response to your comments, we have modified the sub-title in Section 3-7 as follows:

The performance of the QM method by selecting the appropriate distribution fitted on two parts divided based on optimum δ is presented in this section. The best distributions determined for above and below of the selected δ at 22 stations are provided in Table S3. Overall, the Gamma exhibited the best performance for the data below δ for GCMs and observed precipitation (106 times), followed by Weibull (38 times) and Lognormal (32 times). The Gamma was also the best in fitting GCMs and observed data above δ (97 times), followed by Weibull (38 times) and Lognormal (32 times).

**Comment**

For all figures, delete the "The" at the beginning place of the corresponding title.

**Answer**

Thanks for your comment. In response to your comments, we have modified the whole figure in this article.

**Comment**

Line 372: "based on" instead of "based".

**Answer**

We have revised the sub-title of Section 4-3 to read as follows to improve the reader's understanding:

**4.3 Performance in reconstructing precipitation climatology**

**Comment**

Line 418-419: Please ensure the sentence is complete.

**Answer**

Thanks for your comment. In response to your comments, we have modified a sentence in Section 4-4 as follows:

This study compared the extreme values of bias-corrected precipitation using four QM methods based on GEV distribution. The bias corrected GCM precipitation above the 95th percentile are presented in Figure 14. L-moment was used to estimate the GEV parameters of bias-corrected GCMs.

**Comment**

Line 433: Add "in" before "Figure 17". There are many places that the sentences are not complete.

**Answer**

Thanks for your comment. In response to your comments, we have modified a sentence in Section 4-4 as follows:

The obtained results for all the GCMs are presented using boxplots in Figure 15.

**Comment**

Line 435: What do you mean by this sentence?

**Answer**

Thanks for your comment. In response to your comments, we have modified a sentence in Section 4-4 as follows:

This result indicates the better performance of F-DDQM in replicating observed precipitation extremes compared to other bias correction methods.

**Comment**

Line 447: Since you mentioned future projection, how do you determine the δ value for extreme precipitation in the future period using the methods in this study?

**Answer**

A δ in a future period reflects the historical δ. Future precipitation can be estimated based on a traditional distribution method. Furthermore, the projection of future precipitation can also be estimated based on the optimal regression equation.

The better it reflects the extreme precipitation in the historical period, the better it reflects the future precipitation. Therefore, this study has focused on more accurate estimates of extreme precipitation over historical periods.

**Comment**

Line 466 and Line 475: Where are the figures for the performances of different fitted distributions, like the gamma distribution and the Weibull distribution?

**Answer**

Thanks for your comment. We presented the performance for three distributions in the supplementary material as follows:

Table S3. The distribution functions determined at 22 stations based on normalized evaluation metrics depending on upper and lower δ (W: Weibull; G: Gamma; L: Lognormal).

| Station | ACCESS | | CanESM5 | | GISS | | INM | | IPSL | | MPI | | MRI | | Nor | |
|---|---|---|---|---|---|---|---|---|---|---|---|---|---|---|---|---|
| | ~ δ% | δ% ~ | ~ δ% | δ% ~ | ~ δ% | δ% ~ | ~ δ% | δ% ~ | ~ δ% | δ% ~ | ~ δ% | δ% ~ | ~ δ% | δ% ~ | ~ δ% | δ% ~ |
| Gangneung | L | G | G | G | G | L | G | G | G | G | G | G | G | G | G | G |
| Gwangju | G | G | G | G | G | G | L | G | G | L | G | W | G | G | G | W |
| Gunsan | G | G | G | G | G | G | G | G | G | L | G | W | G | L | L | L |
| Daegu | G | G | W | G | G | G | W | G | G | L | G | W | G | G | L | W |
| Daejeon | W | G | G | W | G | W | L | G | L | L | G | W | G | L | L | L |
| Mokpo | W | G | G | G | G | G | L | G | G | G | G | G | G | G | G | L |
| Busan | L | G | W | G | W | W | W | G | W | W | W | W | W | W | W | W |
| Seusan | G | G | L | G | G | G | G | G | W | L | G | W | L | L | L | L |
| Seoul | G | G | W | G | G | G | G | G | L | L | L | W | G | L | G | G |
| Sokcho | G | G | L | G | G | W | G | G | G | W | G | L | G | G | G | L |
| Yeosu | W | G | G | G | W | G | G | W | W | W | W | W | G | G | L | W |
| Ulleungdo | W | G | W | G | G | W | G | G | G | W | G | W | G | G | G | G |
| Ulsan | G | W | G | W | G | W | G | G | G | W | G | L | G | L | G | W |
| Incheon | W | W | W | G | G | G | G | G | L | L | G | L | G | W | W | L |
| Jeonju | G | G | W | G | L | G | L | G | L | G | G | W | L | G | L | L |
| Jeju | G | G | G | G | G | G | G | G | G | G | G | G | G | G | G | G |
| Jinju | G | G | L | W | L | G | G | G | G | W | G | W | G | G | L | G |
| Chuncheon | G | G | W | G | G | G | W | G | G | L | G | W | G | W | L | L |
| Cheongju | W | G | G | G | G | G | W | G | G | W | L | W | G | W | G | L |
| Chupungyeong | W | G | W | W | G | W | G | G | L | L | G | W | L | G | L | L |
| Tongyeong | W | G | W | G | G | G | G | G | L | G | G | G | L | G | G | G |
| Pohang | W | W | W | L | G | W | W | W | W | L | W | L | W | L | W | L |

Table S4. The optimal δ of F-DDQM for daily precipitation determined from 8 CMIP6 GCMs and 22 stations based on normalized evaluation metrics results.

| Station | ACCESS | CanESM5 | GISS | INM | IPSL | MPI | MRI | Nor |
|---|---|---|---|---|---|---|---|---|
| Gangneung | 99% (4.9) | 99% (4.91) | 80% (4.84) | 99% (4.89) | 99% (4.93) | 99% (4.73) | 99% (4.99) | 99% (4.97) |
| Gwangju | 99% (4.98) | 99% (4.97) | 89% (4.99) | 88% (5.0) | 74% (5.0) | 83% (4.96) | 97% (5.0) | 78% (4.96) |
| Gunsan | 99% (4.93) | 99% (4.92) | 87% (5.0) | 87% (4.97) | 76% (5.0) | 85% (4.95) | 78% (5.0) | 76% (4.99) |
| Daegu | 99% (4.95) | 86% (5.0) | 86% (5.0) | 89% (5.0) | 83% (4.99) | 83% (5.0) | 85% (4.97) | 83% (4.96) |
| Daejeon | 83% (4.98) | 96% (4.85) | 83% (4.92) | 91% (5.0) | 77% (4.96) | 86% (5.0) | 80% (4.99) | 76% (5.0) |
| Mokpo | 86% (4.99) | 97% (4.98) | 90% (4.97) | 87% (5.0) | 78% (4.98) | 88% (4.98) | 78% (4.99) | 77% (5.0) |
| Busan | 85% (4.97) | 85% (4.94) | 85% (4.93) | 88% (5.0) | 83% (4.97) | 82% (4.93) | 82% (4.98) | 82% (5.0) |
| Seusan | 87% (4.99) | 89% (4.99) | 90% (4.99) | 91% (4.94) | 77% (5.0) | 81% (4.99) | 80% (4.99) | 77% (4.98) |
| Seoul | 99% (4.94) | 99% (4.75) | 84% (4.93) | 88% (4.96) | 78% (5.0) | 85% (4.99) | 81% (5.0) | 95% (4.95) |
| Sokcho | 80% (4.98) | 84% (4.98) | 81% (4.96) | 87% (5.0) | 80% (5.0) | 80% (5.0) | 85% (4.99) | 79% (4.99) |
| Yeosu | 99% (4.94) | 99% (4.92) | 84% (5.0) | 84% (5.0) | 82% (5.0) | 82% (4.98) | 84% (4.98) | 82% (5.0) |
| Ulleungdo | 99% (4.92) | 86% (5.0) | 73% (4.99) | 84% (4.94) | 73% (4.97) | 73% (5.0) | 79% (4.99) | 98% (5.0) |
| Ulsan | 82% (4.93) | 82% (4.95) | 83% (4.97) | 88% (4.96) | 82% (4.98) | 82% (4.98) | 82% (4.98) | 82% (4.98) |
| Incheon | 83% (4.97) | 98% (4.93) | 87% (4.93) | 88% (5.0) | 81% (5.0) | 79% (5.0) | 83% (4.99) | 79% (5.0) |

| | | | | | | | | |
|---|---|---|---|---|---|---|---|---|
| Jeonju | 84% (4.98) | 97% (4.98) | 84% (5.0) | 89% (5.0) | 97% (4.96) | 85% (4.95) | 74% (4.97) | 74% (4.99) |
| Jeju | 99% (5.0) | 98% (5.0) | 98% (4.99) | 99% (5.0) | 98% (4.98) | 97% (4.86) | 91% (4.95) | 97% (4.99) |
| Jinju | 99% (4.97) | 84% (4.95) | 87% (4.99) | 89% (5.0) | 82% (4.97) | 83% (4.92) | 85% (4.96) | 84% (4.99) |
| Chuncheon | 99% (4.97) | 88% (4.96) | 85% (5.0) | 91% (4.99) | 79% (5.0) | 84% (4.95) | 81% (4.99) | 77% (5.0) |
| Cheongju | 85% (4.98) | 98% (4.97) | 84% (4.97) | 91% (5.0) | 80% (5.0) | 84% (4.96) | 80% (4.98) | 77% (4.99) |
| Chupungyeong | 83% (4.98) | 80% (4.98) | 82% (4.94) | 87% (5.0) | 77% (4.98) | 85% (4.94) | 83% (5.0) | 77% (5.0) |
| Tongyeong | 88% (4.99) | 87% (5.0) | 87% (5.0) | 89% (5.0) | 84% (4.99) | 85% (4.97) | 88% (4.96) | 84% (4.98) |
| Pohang | 82% (4.95) | 83% (4.81) | 82% (4.98) | 83% (4.94) | 81% (4.81) | 79% (4.97) | 82% (4.97) | 81% (4.83) |

Unselected Weibull and lognormal distributions are lower than the values in the table presented.

**Comment**

Line 441-491: In the section of Discussion, the discussion should be strengthened rather than repeating describing the results in the part of results.

**Answer**

Thanks for your comment. In response to your comments, we have modified the Discussion section.

We have focused on the following topics in the discussion section.

Why develop F-DGQM and F-DDQM?

Here, we pointed out the use of specific quantiles and distribution functions, despite using many stations in previous studies. In addition, the importance of selecting an appropriate distribution function was emphasized based on prior research on the combination of distribution functions.

Was the use of F-DGQM and F-DDQM appropriate?

Here, we found that using different deltas for each station was closely related to improving bias correction performance. In addition, the QM method developed in this study performed better than SGQM or DGQM, so we conducted a discussion to secure the legitimacy of the developed method.

Can the proposed methods be able to estimate the extreme precipitation well?

Based on the JSD-KLD results, we found that the extreme precipitation of the two methods proposed here is more similar to the observed data than the existing methods. Therefore, it satisfies bias performance, extreme precipitation estimation, and spatial distribution that must be secured in the historical period.

---

## Author Comment (AC3)

**Reviewer 3**

**Comment**

It is not really appropriate to use monthly precipitation to validate the proposed methods in correcting precipitation extremes from my perspective. For the precipitation extremes, the daily or sub-daily scale precipitation data is required.

**Answer**

Thanks for your comment. We agree with your comments. Therefore, we changed the data to daily precipitation, and it was all reflected in the article.

**Comment**

What is the relationship between F-DDQM and F-DGQM authors proposed? Since the F-DDQM performs best because of the consideration of diverse distribution function, what is the point of the F-DGQM's existence?

**Answer**

Thanks for your comment. In this study, when performing quantile mapping by dividing into two segments, the first goal is to find an appropriate delta because the distribution of precipitation is different for each station.

F-DGQM determines the appropriate delta for distribution by station based on five evaluation metrics. Based on the F-DGQM results, we demonstrated that the dividing point could be different for each station because each station's scale and trend of precipitation were different.

The second goal of this study is to select the distribution in each segment. F-DDQM may outperform F-DGQM in historical performance and emphasizing research creativity. Our study also presented F-DDQM, a method that can achieve both goals.

**Comment**

Segmenting the precipitation series to two fragments by selecting an optimal threshold, is a good idea. But in my opinions, it is not optimal to use QM based on the theoretical distribution function approach for two different sequences. For the non-extreme series, the non-parametric transformation, i.e., interpolation method in Q-Q plot, is able to capture more precipitation

information compared with parametric transformation method and theoretical distribution function method. For the extreme series, when the future precipitation extremes lie outside the domain of historical model data, the simple extrapolation algorithm, such as linear, cubic, and spline interpolation, might lead to great bias. So, in this situation, the theoretical distribution function can be applied due to its advantage of extending the data reasonably.

**Answer**

Thanks for your comment. We agree with your comment. When projecting future precipitation with the suggested method, we applied the selected distribution for the future. For example, if the selected distributions are Weibull and Gamma for the two segments (1%-70%: Weibull; 71%-100%: Gamma) for the historical period, respectively, we used the Weibull distribution to project into the future precipitation below the 70th quantile, and the gamma distribution to project future precipitation of above 71st quantile. These methods can more accurately reflect the distribution of precipitation over historical periods. In addition, the extreme precipitation for the future period can be better than one distribution.

**Comment**

For equation 3, the Fg corresponds the GCM outputs and the Fg-1 corresponds the observed data. I think this is wrong. Different letters subscript should be used.

**Answer**

Thanks for your comment. In response to your comments, we modified the Equation 3 as follows:

$$P_g(t) = F_o^{-1}\big(F_g(P_m(t), \alpha_m, \beta_m), \alpha_o, \beta_o\big) \tag{3}$$

The modified the paragraph as follows:

where $P_g(t)$ denotes the bias-corrected monthly precipitation, $P_m(t)$ represents GCM raw data, $F_o^{-1}$ is the inverse CDF of the observed data to which the gamma function is applied, and $F_g$ is the CDF of the GCM outputs. $\alpha_o$, $\alpha_m$, $\beta_o$ and $\beta_m$ represent shape and scale parameters of observed and GCM simulation, respectively.

**Comment**

I don't know why the authors used the GEV to fit the corrected models data compared with observation. A direct comparison of the empirical distribution functions of the extreme value series seems more appropriate. Additionally, for the extreme series obtained by POT model, GP distribution is generally more appropriate, rather than GEV.

**Answer**

Thanks for your comment. The GEV distribution is popularly used to estimate the extremes. Its performance has also been demonstrated in several studies.

This study did not use GEV to fit the bias-corrected model data. Instead, we compared the corrected precipitation using the GEV distribution using the quantile mapping methods. The results of KLD and JSD showed the difference between bias-corrected data by quantile mapping and observed data.

---

## Author Comment (AC4)

**Reviewer 1**

**Comment**

A few notes on annual rainfall value might help in assessing the typical climate patterns over the study area. Describing trend of rainfall values would be nice. Also, where was the climate data collected? [L91]

**Answer**

Thanks for your comment. In response to your comments, we added sentence in Section 2-1 as follows:

The daily precipitation data was obtained from the Korea Meteorological Administration (KMA) (https://www.weather.go.kr/w/index.do). The annual average precipitation ranges between 1000 mm and 1600 mm. The majority of precipitation occurs in summer.

**Comment**

3.3 Flexible double gamma quantile mapping (F-DGQM) - > bold [L156]

**Answer**

Thanks for your comment. In response to your comments, we modified sub-title in Section 3-3 as follows:

**3.3 Flexible double gamma quantile mapping (F-DGQM)**

**Comment**

I wonder why authors use RMSE to determine delta. For example, several studies use AIC or BIC to find a suitable distribution. However, the authors determined the distribution using only RMSE.

**Answer**

Thanks for your comment. We reselected deltas using normalized values of five metrics instead of RMSE. It is clearly mentioned in the text of the revised manuscript.

AIC and BIC are closer to the optimum as the distribution scale is smaller. Therefore, AIC and BIC may not be suitable for selecting a delta.

Because JSD and KLD evaluated the performance of bias-corrected precipitation, Sections 3-7 should be included in Sections 3-5.

**Answer**

Thanks for your comment. In response to your comments, we modified the Section 3-5 [L216-

L252]

**Comment**

As shown in Figure 5, the most suitable deltas are at both extremes. Authors should add a discussion of the determined delta to section 5 or section 4-1.

**Answer**

Thanks for your comment. The delta selection results changed after we changed the data to daily precipitation. Most of the deltas were found to be located in the middle quantiles, described in Section 5 as follows:

The  $\delta$  of F-DGQM was different at different stations. The middle quantiles were the most selected  $\delta$ . It means that the suitable  $\delta$  at a station depends on the scale and shape of the GCM precipitation distribution. Therefore, the determination of  $\delta$  can influence the estimate of extreme precipitation. It was reasonable to use normalized evaluation metrics to determine two segments.

The F-DDQM showed better performance than F-DGQM because of considering three probability distributions for two individual segments. In F-DDQM, the  $\delta$  was mainly selected in the median percentile (80%-89%). Further, the selected  $\delta$  at some stations were specific to 99% and 82%. The Gamma distribution performed the best for two individual segments, followed by Weibull. The results prove that the Lognormal PDF is not proper in analyzing the daily precipitation of South Korea. The performance of F-DDQM was higher than F-DGQM in all evaluation metrics. Furthermore, the performance improvement using F-DDQM was more than F-DGQM at all stations.

**Comment**

As shown in Figure 5, the most suitable deltas are at both extremes. Authors should add a discussion of the determined delta to section 5 or section 4-1.

**Answer**

Thanks for your comments. We have added the following sentences to Sections 4-1 and 4-2 in response to your comments:

The modified Section 4-1 is as follows:

These results showed that F-DGQM performed better than other methods for below  $\delta$  precipitation because the MD was less sensitive than NSE to extreme values.

The modified Section 4-2 is as follows:

The results indicate better performance of F-DDQM than the other methods for below  $\delta$  precipitation.

**Comment**

Figure 6: bais - > bias

**Answer**

Thanks for your comments. In response to your comment, we modified Figure 5 as follows:

Figure 6. Performance of three QM methods in correcting GCM simulated daily precipitation bias at 22 stations in South Korea.

In the scatter plot of Figure 8, it isn't easy to discern the difference between F-DGQM and DGQM. Therefore, the authors should remove figure 8 as these results have already demonstrated a difference with the evaluation indices. It would also be nice to present an annual time series figure, but it is unnecessary to add it.

**Answer**

Thanks for your comment. We have removed the scatter plots in sections 4-1 and 4-2 in response to your comments.

**Comment**

My comment on Figure 10 is like Figure 5. Authors need to improve their results.

**Answer**